# Origins of glycan selectivity in streptococcal Siglec-like adhesins suggest mechanisms of receptor adaptation

Barbara A. Bensing[1,2,16], Haley E. Stubbs[3,16], Rupesh Agarwal [4,5], Izumi Yamakawa[6,14], Kelvin Luong[6], Kemal Solakyildirim[7,8], Hai Yu [8], Azadeh Hadadianpour[9], Manuel A. Castro[10], Kevin P. Fialkowski [6,15], KeAndreya M. Morrison[11], Zdzislaw Wawrzak[12], Xi Chen[8], Carlito B. Lebrilla [8], Jerome Baudry[13], Jeremy C. Smith[4,5], Paul M. Sullam[1,2] & T. M. Iverson [6,10 ✉]

Bacterial binding to host receptors underlies both commensalism and pathogenesis. Many streptococci adhere to protein-attached carbohydrates expressed on cell surfaces using Siglec-like binding regions (SLBRs). The precise glycan repertoire recognized may dictate whether the organism is a strict commensal versus a pathogen. However, it is currently not clear what drives receptor selectivity. Here, we use five representative SLBRs and identify regions of the receptor binding site that are hypervariable in sequence and structure. We show that these regions control the identity of the preferred carbohydrate ligand using chimeragenesis and single amino acid substitutions. We further evaluate how the identity of the preferred ligand affects the interaction with glycoprotein receptors in human saliva and plasma samples. As point mutations can change the preferred human receptor, these studies suggest how streptococci may adapt to changes in the environmental glycan repertoire.

[1] Division of Infectious Diseases, Veterans Affairs Medical Center, Department of Medicine, University of California, San Francisco, CA, USA. [2] the Northern California Institute for Research and Education, San Francisco, CA 94121, USA. [3] Graduate Program in Chemical and Physical Biology, Vanderbilt University, Nashville, TN 37232, USA. [4] University of Tennessee/Oak Ridge National Laboratory, Center for Molecular Biophysics, Biosciences Division, Oak Ridge National Laboratory, Oak Ridge, TN 37831-6309, USA. [5] Department of Biochemistry and Cellular and Molecular Biology, University of Tennessee, Knoxville, TN 37996, USA. [6] Department of Pharmacology, Vanderbilt University, Nashville, TN 37232, USA. [7] Department of Chemistry, Erzincan Binali Yildirim University, Erzincan 24100, Turkey. [8] Department of Chemistry, University of California, Davis, CA 95616, USA. [9] Department of Microbiology, Pathology, and Immunology, Vanderbilt University, Nashville, TN 37232, USA. [10] Department of Biochemistry, Vanderbilt University, Nashville, TN 37232, USA. [11] Department of Pharmacology, School of Graduate Studies and Research, Meharry Medical College, Nashville, TN 37208, USA. [12] LS-CAT Synchrotron Research Center, Northwestern University, Argonne, IL 60439, USA. [13] Department of Biological Sciences, The University of Alabama in Huntsville, Huntsville, AL 35899, USA. [14] Present address: School of Nursing, Belmont University, Nashville, TN 37212, USA. [15] Present address: College of Medicine, University of Arkansas for Medical Sciences, Little Rock, AR 72205, USA. [16] These authors contributed equally: Barbara A. Bensing, Haley E. Stubbs. ✉email: tina.iverson@vanderbilt.edu

Selection among many possible host receptors determines whether a bacterium can adhere to a preferred anatomical niche or can infect a particular host[1,2]. Streptococci and staphylococci are among the organisms that use host-associated carbohydrates as receptors; they may specifically bind to sialic acid-containing glycans (sialoglycans; Fig. 1). As an example, human O-linked glycosylated proteins commonly contain a terminal α2-3-linked sialic acid-galactose disaccharide, (Neu5Acα2-3Gal). Additional forms of sialic acid and alternative linkages are found in animal sialoglycans[3,4].

Neu5Acα2-3Gal is present on the human salivary mucin MUC7[5–7], on several glycoproteins in blood plasma[8], and on surface platelet proteins[9,10]. Bacterial binding to glycoproteins terminating with Neu5Acα2-3Gal may therefore allow colonization of the oral cavity as a commensal. In animal models, sialoglycan binding is also implicated in the persistence of these organisms in the bloodstream as an endovascular pathogen[11–14], although it is not known whether all streptococci can act as pathogens.

Siglec-like binding regions (SLBRs) are among the streptococcal adhesive modules that bind sialoglycans. SLBRs are usually found within the context of serine-rich repeat proteins, which form fibrils extending from the bacterial surface. SLBRs contain two adjacent modules: a Siglec domain, which shares some features with mammalian Siglecs, and a Unique domain[13] with no close homologs outside of the family. The Siglec domain contains a ΦTRX sequence motif[15] that recognizes Neu5Acα2-3Gal in the context of larger glycans. Reported mutagenesis of the ΦTRX motif demonstrates its importance in sialoglycan binding[5,13,16] and in endovascular disease in animal models[13]. This has motivated the development of compounds that bind the ΦTRX motif as a potential therapy for human endovascular infections caused by these organisms[17,18].

SLBRs display a range of selectivity. Some SLBRs bind selectively to the α2-3-linked trisaccharide sialyl-T antigen (sTa, Neu5Acα2-3Galβ1-3GalNAc; Fig. 1a)[5,19]. Others have intermediate selectivity and bind to a small number of closely related glycans[5,19]. Still others can bind to a broad range of sialoglycans and do not distinguish between related structures[5,19]. The binding profile of these SLBRs is likely adapted to the host display of sialoglycans. In the oral cavity for example, the display of sialylated O-glycans on MUC7 varies between individuals, making it possible that the binding preferences of the SLBRs reflect the specific glycosylation display of an individual[5–7,20,21]. The binding profile can also affect virulence; streptococci containing SLBRs that preferentially bind to sTa in vitro exhibit higher pathogenicity in an animal model of endocarditis[22].

Despite the importance of the sialoglycan binding profile in the interaction between streptococci and host[22], the sequence determinants that underlie glycan selectivity are not currently clear. Here, we determine the molecular basis for glycan selectivity of a phylogenetically-informed library of SLBRs. We test our predictions for selectivity by engineering the binding spectrum of selected SLBRs and assessing host receptor switching in human saliva and plasma glycoproteins. Collectively, these studies improve our understanding of the glycan selectivity that underlies commensalism and pathogenesis. In addition, they suggest how these bacteria may adapt to host sialoglycan repertoires.

## Results

### Selection of SLBRs for study
Starting with SLBRs with at least some previously-reported selectivity, we correlated selectivity with phylogeny (Fig. 2)[5,8,19,23]. Our initial trees contained two major branches. This identified that evolutionary relatedness of SLBRs is a moderate, but not strong, predictor of glycan selectivity. Most SLBRs of the first major branch of the tree (blue in Fig. 2) are broadly-selective and recognize two or more related tri-, tetra-, or hexasaccharides (see examples in Fig. 1). However, sequence similarly does not clearly predict the preferred glycan[5,8,19,23]. In contrast, characterized SLBRs of the second major branch (green in Fig. 2) are selective for sTa (Fig. 1a)[5,8,19,23].

To understand selectivity of these SLBRs for human glycans, we chose comparators from each branch for detailed study. From the first branch of the tree (blue in Fig. 2), we selected the SLBRs of the Hsa adhesin from S. gordonii strain Challis (termed SLBR$_{Hsa}$) and the equivalent SLBRs from Streptococcus sanguinis strain SK678 (SLBR$_{SK678}$) and Streptococcus gordonii strain UB10712 (SLBR$_{UB10712}$; see footnote). Although these three SLBRs are >80% identical in amino acid sequence, when they were tested with arrays containing 49 sialoglycans, they exhibited distinct binding profiles[5,19]. SLBR$_{Hsa}$ was quite broadly selective and bound to a range of α2-3-linked sialoglycans, but not to the corresponding fucosylated derivatives[5,19]. In comparison, SLBR$_{UB10712}$ and SLBR$_{SK678}$ were more narrowly selective, although both bound to multiple sialoglycan ligands. Specifically, SLBR$_{UB10712}$ bound strongly to 3'-sialyl-N-acetyllactosamine (3'sLn; Neu5Acα2-3Galβ1-4GlcNAcβ, Fig. 1b) and a small range of related structures[5], while SLBR$_{SK678}$ bound to only two of the glycans on this array, 3'sLn and 6-O-sulfo-sialyl Lewis X (6S-sLe$^X$; Neu5Acα2-3Galβ1-4(Fucα1-3)GlcNAc6Sβ, Fig. 1c)[5]. In summary, all three of these SLBRs bind multiple ligands with promiscuity following SLBR$_{Hsa}$ > SLBR$_{UB10712}$ > SLBR$_{SK678}$.

The second major branch of the evolutionary tree (green in Fig. 2) includes the well-characterized SLBR$_{GspB}$ from S. gordonii strain M99[7,9,13,24,25]. SLBR$_{GspB}$ exhibits narrow specificity for the sTa trisaccharide, as have other previously-characterized members of this evolutionary branch[5,8,19,23,24]. The binding results for GST-SLBR$_{GspB}$ with sTa, 3'sLn, and sialyl Lewis$^C$ (sLe$^C$, Neu5Acα2-3Galβ1-3GlcNAc) (Fig. 1d) were recapitulated here by ELISA showing concentration-dependent binding (Supplementary Fig. 1a).

In seeking comparators of SLBR$_{GspB}$, we evaluated close homologs for their binding spectrum. We identified that a previously-uncharacterized SLBR from Streptococcus sanguinis strain SK150 (termed SLBR$_{SK150}$) displays 62% identity to SLBR$_{GspB}$ but exhibits a distinct binding profile (Supplementary Fig. 1b). In short, there was modest binding to each of the three trisaccharides, i.e., sTa, 3'sLn, and sLe$^C$, but little detectable binding to any of the tetrasaccharides (i.e., 6S-sLe$^X$ (Fig. 1c), sialyl Lewis X (sLe$^X$; Neu5Acα2-3Galβ1-4(Fucα1-3)GlcNAcβ; Fig. 1d), and sialyl Lewis$^A$ (sLe$^A$; Neu5Acα2-3Galβ1-3(Fucα1-4)GlcNAcβ; Fig. 1e)) (Supplementary Fig. 1b). The high sequence similarity and distinct binding properties of SLBR$_{GspB}$ and SLBR$_{SK150}$ make these good comparators for understanding selectivity.

### Structural basis for recognition of sialoglycan elaborations
To reveal how similar SLBRs could include or exclude different sialoglycans, we determined crystal structures of these five SLBRs at resolutions between 1.0 Å and 1.7 Å (Supplementary Tables 1, 2, Fig. 3, and Supplementary Fig. 2). This included a structure of SLBR$_{GspB}$ with improved resolution as compared to a previous report[13]. In each structure, the N-terminal Siglec domain is organized around a V-set variation of the Ig fold (Fig. 3), named for its discovery in antibody variable domains[26]. The C-terminal Unique domain of the SLBRs displays a fold that has only been observed in other members of this family (Supplementary Fig. 2).

We next evaluated how these SLBRs interact with preferred versus disfavored ligands. Only the crystallization conditions for SLBR$_{Hsa}$ and the isolated Siglec domain of SLBR$_{GspB}$

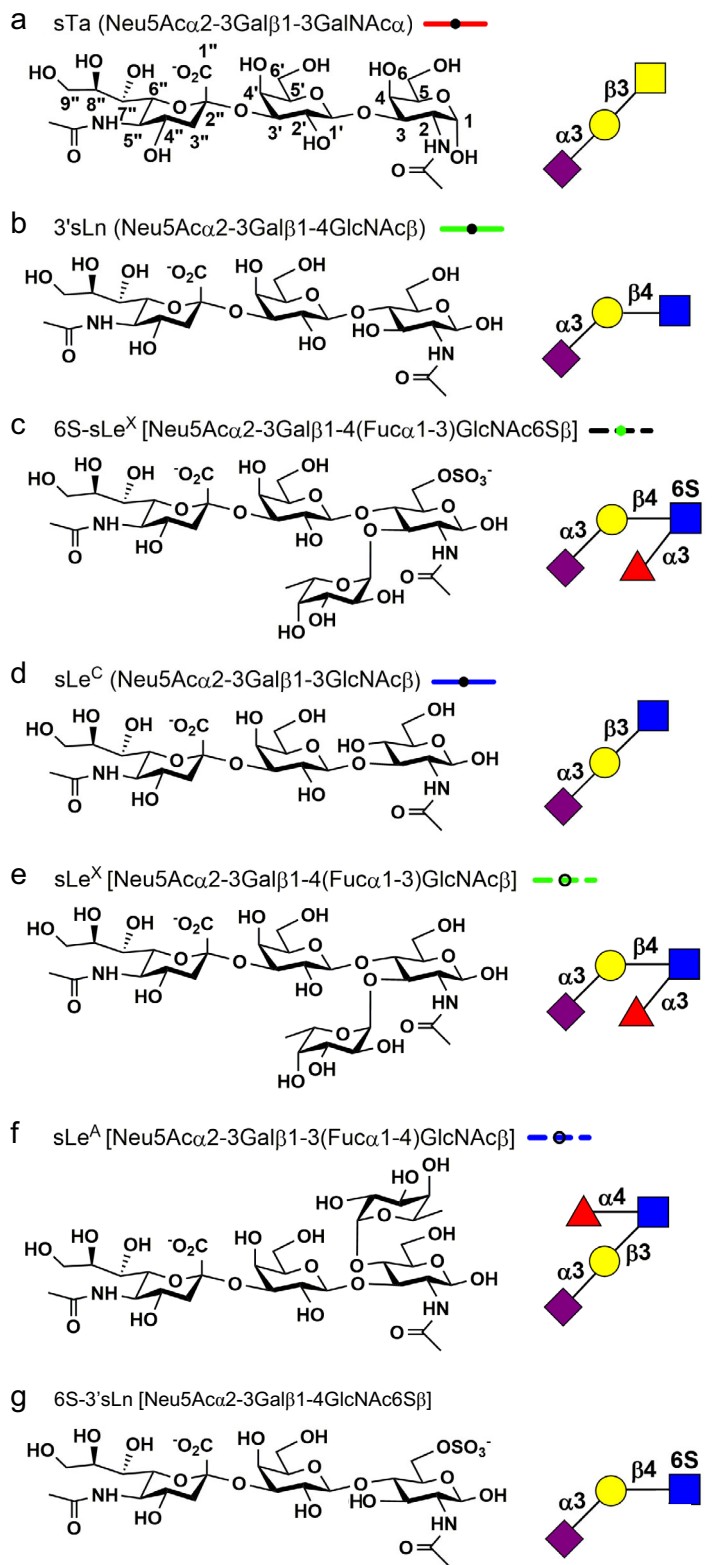

**Fig. 1 Sialoglycans used in this study. a** sialyl-T antigen, **b** sialyllactosamine, **c** 6S-sialyl Lewis X, **d** sialyl Lewis$^{C}$, **e** sialyl Lewis X, **f** sialyl Lewis A, **g** 6S-sialyllactosamine. The chemical structure of each indicated sialoglycan is shown on the left with the symbolic representation shown on the right. The line style used for all dose response curves is shown to the right of each name.

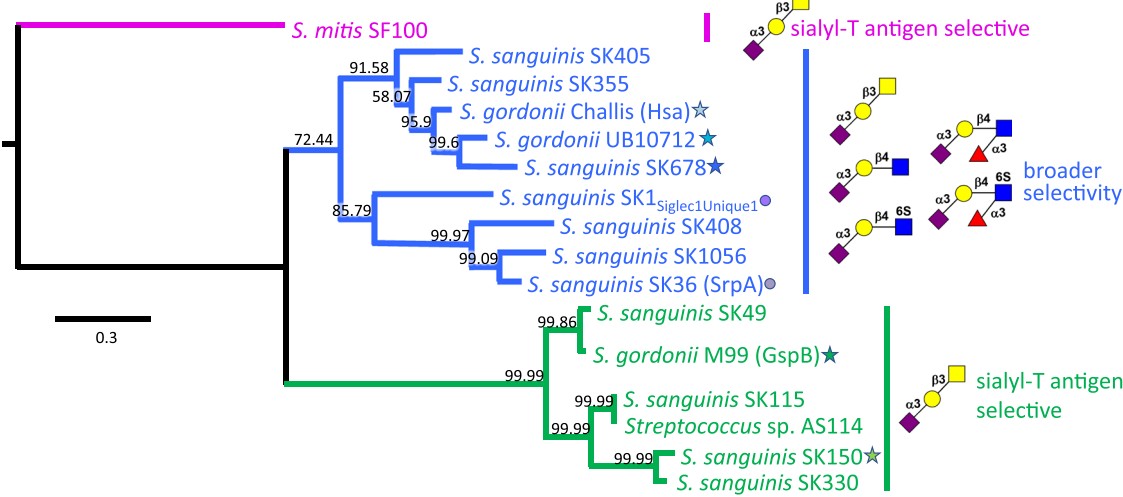

**Fig. 2 Phylogeny of select bacterial SLBRs.** Phylogenetic analysis of the SLBRs from the indicated strains comprising the tandem Siglec and Unique domains reveals three distinct subgroups. Glycans are depicted using standard symbol nomenclature, with linkage designations shown as numbers and the 6S elaborations shown in red. Characterized SLBR$_{Hsa}$-like SLBRs (*blue*) bind to two or more of the indicated sialoglycans; the previously-characterized SLBR$_{GspB}$-like SLBRs (*green*) exhibited narrow selectivity for sialyl-T antigen. The tree is rooted using the distantly related *S. mitis* SLBR$_{SF100}$ (*magenta*). SLBRs investigated here are highlighted with a star with the color family reflect the branch of tree; later figure panels comparing properties of these SLBRs follow this coloring. The structure and ligand binding properties of SLBR$_{SrpA}$ and SLBR$_{SK1}$ are highlighted with circles as they have previously been reported[13,15,16,23] and are used as comparators in this report. The scale bar indicates the average number of nucleotide substitutions per site, and the numbers on each branch represent the confidence of inferred tree topology.

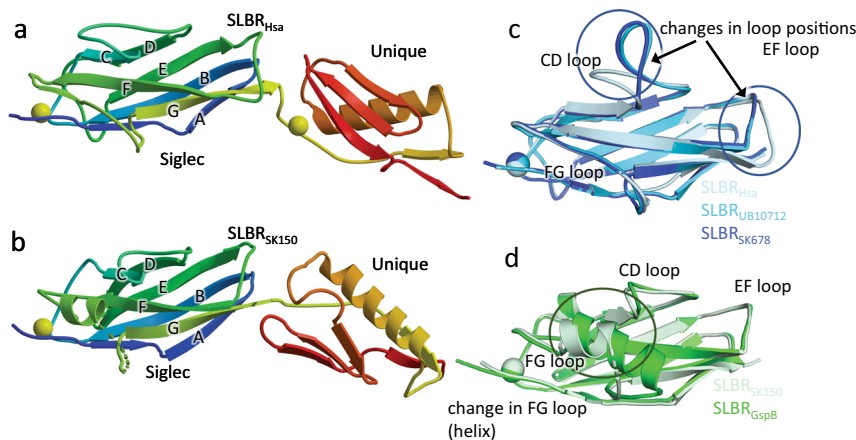

**Fig. 3 Structural differences between SLBR$_{Hsa}$, SLBR$_{UB10712}$, and SLBR$_{SK678}$.** Ribbon diagrams of **a** SLBR$_{Hsa}$ and **b** SLBR$_{SK150}$ with the N-terminus in *blue* and the C-terminus in *red*. Ions are shown as spheres. **c** Overlay of the Siglec domain from the SLBR$_{Hsa}$ (*gray-blue*) SLBR$_{UB10712}$ (*cyan*) and SLBR$_{SK678}$ (*blue*). The color of each SLBR is the same as the color of the stars for each strain in Fig. 2. **d** Overlay of the Siglec domain in the SLBR$_{GpsB}$ (*green*) and SLBR$_{SK150}$ (*light green*). The CD, EF, and FG loops are highlighted. These are poorly conserved in sequence and length (Supplementary Fig. 4) and display significant structural variability.

(SLBR$_{GspB-Siglec}$) supported sialoglycan binding (Supplementary Table 3). For SLBR$_{Hsa}$, this included structures from crystals soaked with the high-affinity ligands sTa (Figs. 1a, 4a, and Supplementary Fig. 3) and sLe$^C$ (Figs. 1d, 4b, and Supplementary Fig. 4), the intermediate-affinity ligand 3'sLn (Figs. 1b, 4c, and Supplementary Fig. 5), and the low-affinity ligand 6S-sLe$^X$ (Figs. 1c, 4d, and Supplementary Fig. 6). The resolution ranged from 1.3 Å to 2.4 Å and the diffraction quality loosely correlated with ligand affinity (Supplementary Table 3). Cocrystals of SLBR$_{GspB-Siglec}$ with sTa diffracted to 1.25 Å resolution and the resultant maps contained unambiguous electron density for the sTa ligand (Fig. 4e, Supplementary Fig. 7, and Supplementary Table 3). This structure is superior to a reported structure of SLBR$_{GspB}$ with sTa, where the low occupancy of the ligand made its assignment ambiguous[13].

The sialoglycan-bound structures of SLBR$_{Hsa}$ and SLBR$_{GspB-Siglec}$ identifies that the sialic acid of all glycans binds above the ΦTRX motif in a similar way. This suggests that while the ΦTRX motif is important for binding, it does not select between potential ligands. More careful comparison suggests that the distinct selectivity may originate from three loops of the V-set Ig fold that surround the sialoglycan binding site: the CD loop (SLBR$_{Hsa}$$^{284-296}$ or SLBR$_{GspB}$$^{440-453}$), the EF loop (SLBR$_{Hsa}$$^{330-336}$ or SLBR$_{GspB}$$^{475-481}$), and the FG loop (SLBR$_{Hsa}$$^{352-364}$ or SLBR$_{GspB}$$^{499-511}$) (Fig. 4 and Supplementary Fig. 8). Variation of both sequence and structure of SLBRs disproportionately maps to these loops (Supplementary Figs. 8 and 9). Moreover, temperature factor analysis suggests that these loops have high flexibility in the absence of ligand (Supplementary Fig. 10). Finally, molecular dynamics (MD) simulations of unliganded SLBR$_{Hsa}$ and SLBR$_{GspB}$ predict that these

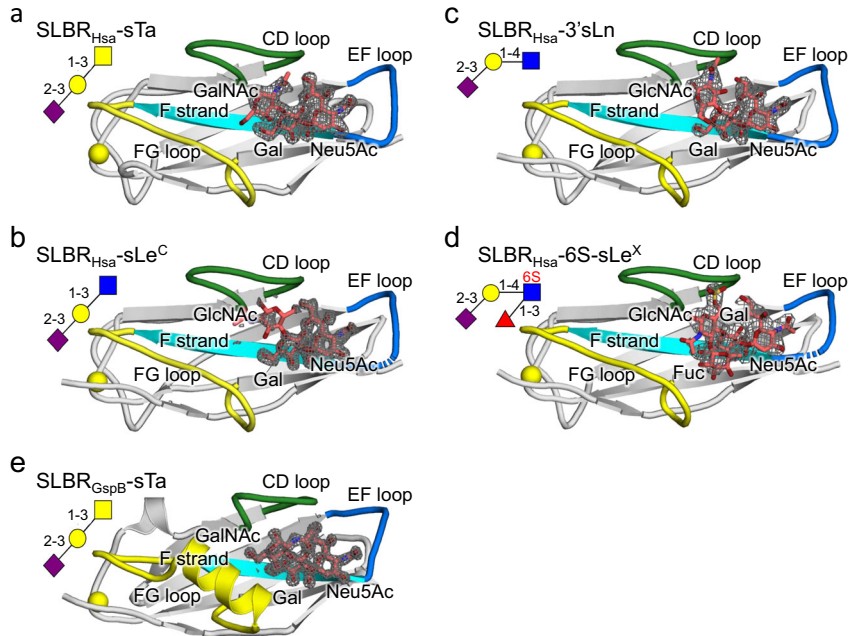

**Fig. 4 Sialoglycans bound to SLBR_Hsa and SLBR_GspB.** SLBR_Hsa bound to sialoglycans **a** sTa, **b** sLe^C, **c** 3'sLn and **d** 6S-sLe^X. **e** SLBR_GspB-Siglec bound to sTa. In each panel, the SLBR is shown as a cartoon with the CD, EF, and FG selectivity loops colored in *green*, *blue*, and *yellow* respectively. The F strand contains the conserved YTRY motif and is shown in *cyan*. Ions are shown in *yellow* spheres. Carbon atoms of each sialoglycan are colored *salmon* with nitrogen shown in *blue* and oxygen in *red*. |F_o| − |F_c| difference electron density calculated after removing the sialoglycan and performing three rounds of refinement in Phenix[55] are shown in gray mesh and contoured at 3σ. The standard depiction for each carbohydrate is shown in the upper left, with linkages indicated.

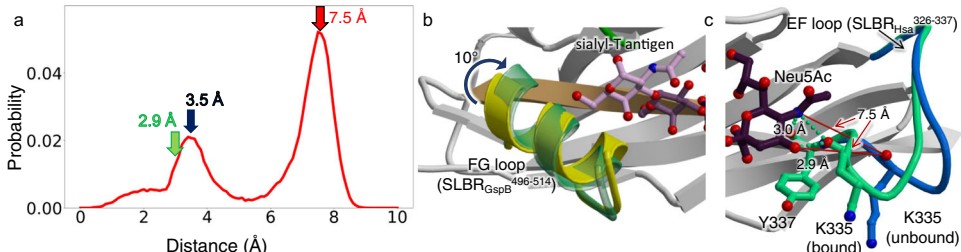

**Fig. 5 Conformations associated with SLBRs bound to sialoglycans. a** Probability of distance distribution between the position of the Neu5Ac O4-hydroxyl in sTa and the SLBR_Hsa^K335 backbone carbonyl, as calculated by MD simulations. A bimodal distribution of distances exhibit maxima at 7.5 Å (*red arrow*), which reflects the unliganded crystal structure, and at 3.5 Å (*navy arrow*), which approaches the liganded crystal structure. The formation of the hydrogen-bond between the SLBR_Hsa^K335 carbonyl and Neu5Ac likely shifts the conformational equilibrium to a pose that supports the 2.9 Å distance (*light green* arrow) observed in the bound state. **b** The FG loop of SLBR_GspB-Siglec rotates 10° upon sTa binding. The position in the unbound structure is shown in *yellow* and the position in the bound structure is shown in *light green*. **c** The EF loop of SLBR_Hsa adjusts to promote formation of hydrogen-bonding interactions between SLBR_Hsa^K335 and the Neu5Ac of sTa. The position of this loop in the unbound structure is shown in *blue*, and the position occupied in the bound structure is shown in *light green*. The distance between the SLBR_Hsa^K335 backbone carbonyl and the position of the Neu5Ac O4-hydroxyl of the unliganded state are shown in *red lines* and match the 7.5 Å distance calculated by MD simulations (panel **a**). The distance between the SLBR_Hsa^K335 backbone carbonyl and the position of the Neu5Ac O4-hydroxyl is shown in *light green dots*.

loops exhibit considerably more flexibility than other parts of the protein (Fig. 5a and Supplementary Fig. 11). The MD further suggests that these loops sample the ligand-bound form even in the absence of sialoglycan. This supports a conformational selection mechanism, where structural change of the protein precedes binding of ligand[27]. The timing of ligand-associated conformational changes in enzymes affects fidelity[28] and may similarly contribute to ligand selectivity in binding proteins.

Distinct loops in SLBR_Hsa and SLBR_GspB showed the largest conformational differences between the unbound and sialoglycan-bound structures. This provides the first hints into how narrow- versus broad-selectivity is conferred in this family. In the sTa-bound structure of SLBR_GspB-Siglec, the helix of the FG loop is rotated 10° as compared to the unliganded conformation. This rotation results in a maximum physical displacement of 1.3 Å (Fig. 5b), which optimizes contacts to the GalNAc of sTa. Mechanistically, this would be consistent with the conserved region of the glycan first interacting with a relatively pre-formed binding pocket comprised of the CD and EF loops prior to interaction with the FG loop.

In SLBR_Hsa, the conformation of the FG loop is similar in the presence and absence of glycan. Instead, comparing costructure determined with sTa with the costructure determined with sLe^C shows that the position of the EF loop differs by 5.9 Å (Fig. 5c). This allows the SLBR_Hsa^K335 carbonyl to form hydrogen-bonding interactions to the invariant portion of the sialoglycans, i.e., the terminal Neu5Acα2-3Gal. In costructures determined with lower-affinity ligands, i.e., 3'sLn or 6S-sLe^X, this loop is not associated with clear electron density. This may result from crystal contacts to the EF loop that stabilize its position in the unliganded pose,

resulting in a mixture of open and closed conformations (Supplementary Fig. 12). Comparison of the EF loop positions in the various crystal structures (Figs. 3b, 4, and Supplementary Fig. 13a) with the positions calculated by the MD simulations (Fig. 5c and Supplementary Fig. 11) suggests that closed conformation of the EF loop in the sTa and 3'sLn-bound crystal structures is likely the lowest energy state (Supplementary Fig. 11). Mechanistically, this suggests that for SLBR$_{Hsa}$, the variable, sub-terminal region of a sialoglycan ligand would first interact with the CD and FG loops. The ligand could then adjust in global position to optimize hydrogen-bonding interactions. The flexibility of the EF loop could then adapt to a range of different orientations of bound sialoglycan. This would be expected to promote broad selectivity. Thus, the location of inherent protein flexibility may define whether an SLBR is narrowly- versus broadly-selective.

To further evaluate how the broadly-selective SLBR$_{Hsa}$ could select for particular sialoglycans, we compared the binding positions of strong, intermediate, and weak ligands (Supplementary Fig. 13). In the strong and intermediate ligands, the invariant Neu5Acα2-3 Gal effectively superimposes (Supplementary Fig. 13a, b) and has similar hydrogen bonds. Differences in the SLBR-ligand interactions predominantly map to the variable third sugar of the glycan (Supplementary Fig. 13c–f). Biding strength may therefore be related to these interactions. In contrast, the global binding position of the weak ligand 6S-sLe$^X$ is shifted as compared with all other ligands (Fig. 4d and Supplementary Fig. 13b, f). This affects the hydrogen bonds along the entirety of the ligand.

6S-sLe$^X$ is both α1,3-fucosylated and O-sulfated at the C6 (6S) of the GlcNAc, modifications that are absent in the strong SLBR$_{Hsa}$ ligands (Fig. 6c). The evaluation of the interactions between these groups and SLBR$_{Hsa}$ suggests how related SLBRs include or exclude these elaborations. In considering how the α1,3-fucose in glycans such as sLe$^X$ and 6S-sLe$^X$ is excluded from SLBR$_{Hsa}$, our analysis suggests that the β-branching of SLBR$_{Hsa}$$^{D356}$ on the FG loop disfavors the binding of a fucosylated glycan (Supplementary Fig. 13c–f). MD simulations also indicate that the FG loop does not sample a position that allows an extra fucose or other large elaboration at this position (Supplementary Fig. 11). This is consistent with the crystal structure, which shows that the loop position does not allow 6S-sLe$^X$ to sit optimally in the sialoglycan binding site.

In considering how a 6 S group might be included or excluded, the structure reveals that SLBR$_{Hsa}$$^{E286}$ of the CD loop contacts the sulfate of 6S-sLe$^X$. This does not exclude a 6S group per se, but both are negatively charged. The structure suggests that an unknown ligand, possibly a component of the buffer, binds near this site to bridge the interaction (Fig. 4d and Supplementary Fig. 13f). Taken together, these structural and computational analyses show that steric and electrostatic interactions of the broadly selective SLBR$_{Hsa}$ exclude specific structural additions to the glycan ligands.

**The CD, EF, and FG loops determine SLBR selectivity.** Because structural studies suggest that the combined action of the CD, EF, and FG loops allow SLBRs to select between ligands, we developed chimeras with the backbone of one SLBR and the loops of a closely-related SLBR. We first replaced the CD, EF, and FG loops of SLBR$_{SK678}$ and SLBR$_{UB10712}$ with the equivalent loops from SLBR$_{Hsa}$ to create the SLBR$_{SK678}$$^{Hsa-loops}$ and SLBR$_{UB10712}$$^{Hsa-loops}$ chimeras. MD simulations would suggest that the loops retain the structure found within the parent SLBR$_{Hsa}$ (Supplementary Fig. 14). Using physiologically-relevant sialoglycans[5,8,19], we measured binding to parent and chimeric SLBRs in ELISAs (Fig. 6a–e). We found that these chimeras

bound glycans strongly and had a sialoglycan-binding preference that closely resembled SLBR$_{Hsa}$ rather than the parent SLBR (Fig. 6f and Supplementary Table 4). This change in selectivity occurred via both a gain-of-function that promoted binding to sTa and a loss-of-function that decreased binding to α1,3-fucosylated and O-sulfated sialoglycans. This change of binding spectrum confirms that a major determinant of selectivity in these SLBRs is the loops that surround the ligand-binding pocket.

We next assessed the contributions of each loop to selectivity (Supplementary Fig. 15). Substitution of the EF loop of SLBR$_{SK678}$ with the EF loop from SLBR$_{Hsa}$ resulted in increased binding to sTa, sLe$^C$, sLe$^X$, and 6S-sLe$^X$ (Supplementary Fig. 15). This result is consistent with the structural prediction that a SLBR with a flexible EF loop can potentially accommodate a greater range of ligands.

In contrast, substitution of the CD or FG loops altered the identity of the preferred ligands. The altered selectivity of these chimeras involved a combination of reduced binding to some sialoglycans and increased binding to others, i.e., both a loss-of-function and a gain-of-function. For example, both SLBR$_{SK678}$$^{Hsa-FG-loop}$ and SLBR$_{UB10712}$$^{Hsa-FG-loop}$ exhibited decreased binding to the fucosylated ligands sLe$^X$ and 6S-sLe$^X$ while SLBR$_{UB10712}$$^{Hsa-FG-loop}$ also increased binding to sTa (Supplementary Fig. 14a, b). This is consistent with the crystallographic interpretation that SLBR$_{Hsa}$$^{D356}$ on the FG loop restricts accommodation of Fucα1-3GlcNAc.

The single-loop chimeras also suggest synergy between these three selectivity loops. For example, the substantial decrease in binding of SLBR$_{SK678}$$^{Hsa-CD-loop}$ to 6S-sLe$^X$ (Supplementary Fig. 15b) is consistent with a proposal that the binding of 6S-ligands is controlled by the CD loop. However, the SLBR$_{UB10712}$$^{Hsa-CD-loop}$ chimera retains robust binding to 6S-sLe$^X$ (Supplementary Fig. 15a) suggesting that the other loops moderate the effects.

We next turned to SLBR$_{GspB}$ and SLBR$_{SK150}$, which both bind sTa preferentially (Supplementary Fig. 1). Here, we substituted the loops of SLBR$_{SK150}$ into SLBR$_{GspB}$ and assessed the binding to sTa and 3'sLn, which are the ligands with the highest affinity for SLBR$_{SK150}$. In contrast to the results observed with SLBR$_{Hsa}$ and its close homologs, substitution of the EF loop of SLBR$_{SK150}$ into SLBR$_{GspB}$ had little impact (Supplementary Fig. 15c). In all remaining chimeras, there was little detectable binding to sTa or 3'sLn (Supplementary Fig. 15c). To determine whether protein misfolding may be a contributing factor in variants with loss of binding, we used size exclusion chromatography (Supplementary Fig. 16a–c), which can distinguish between folded and mis-folded SLBRs[23]. The chromatogram of the SLBR$_{GspB}$$^{SK150-loops}$ showed a monodisperse peak with little aggregation, indicating that loss of binding in this case was not due to misfolding. However, the chromatograms of the SLBR$_{GspB}$$^{SK150-CD-loops}$ and SLBR$_{GspB}$$^{SK15-FG-loops}$ chimeras showed significant levels of protein aggregates and break-down products, indicating that misfolding may contribute to loss of binding for these two variants.

The ability to develop functional chimeras for the three SLBR$_{Hsa}$-like adhesins, but not the two SLBR$_{GspB}$-like adhesins, might be explained in several ways. First, the broadly-selective scaffolds of SLBR$_{Hsa}$, SLBR$_{SK678}$, and SLBR$_{UB10712}$ may have more plasticity, allowing these to better accommodate non-native loops. Conversely, the broadly-selective SLBRs may contain somewhat more flexible loops that more easily adjust to the non-native scaffold. Finally, the sequence identity between SLBR$_{Hsa}$, SLBR$_{SK678}$, and SLBR$_{UB10712}$ is higher than that between SLBR$_{GspB}$ and SLBR$_{SK150}$, allowing a better fit between the scaffold and chimeric loops in the SLBR$_{Hsa}$-like proteins. To better understand why SLBR$_{Hsa}$-like proteins were more mutable, we leveraged our crystal structure of SLBR$_{GspB}$ in complex with sTa (Fig. 4e) and identified that SLBR$_{GspB}$$^{L442}$ and SLBR$_{GspB}$$^{Y443}$ closely approach the GalNAc

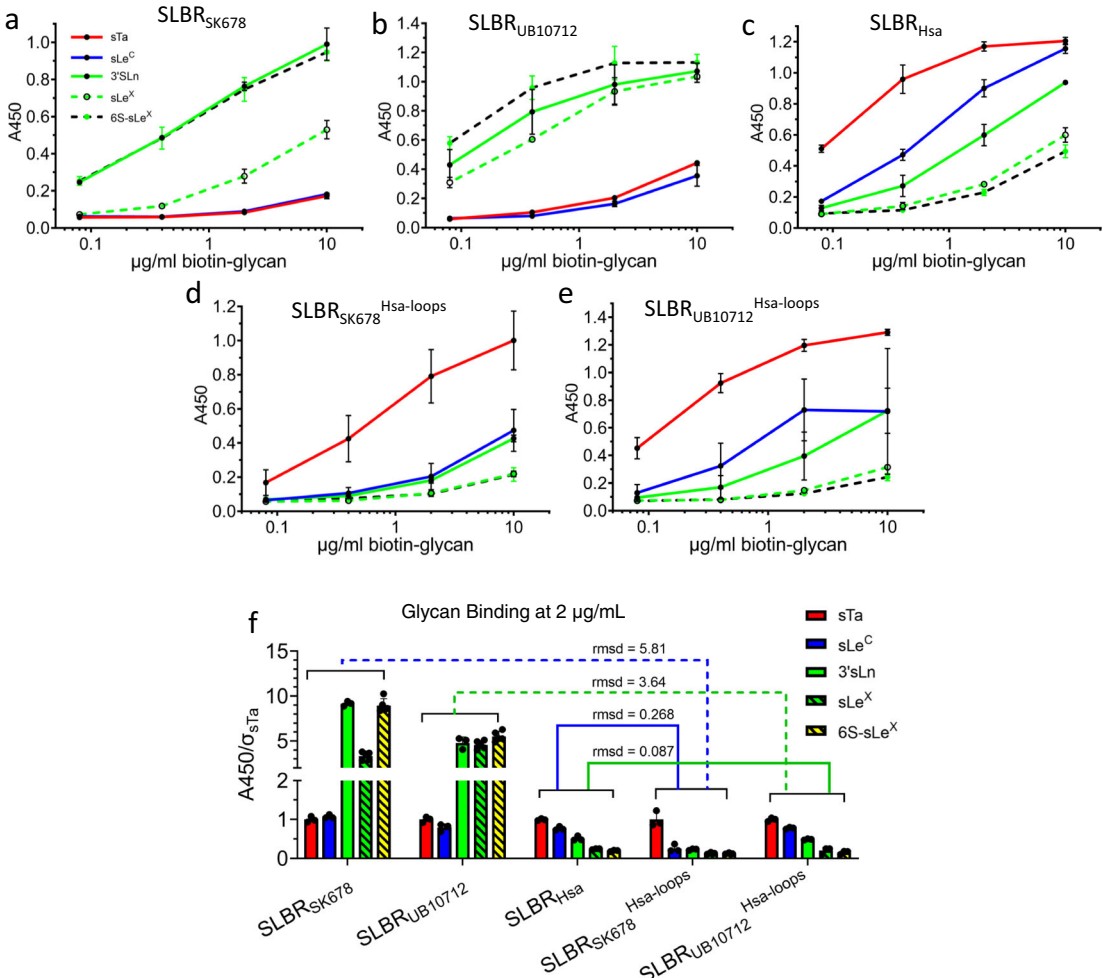

**Fig. 6 Chimeragenesis of SLBR_Hsa and its close homologs.** Dose-response curves of **a** wild-type GST-SLBR_SK678, **b** wild-type GST-SLBR_UB10712, and **c** wild-type GST-SLBR_Hsa to five selected ligands. Dose-response curves of the chimeras **d** GST-SLBR_SK678^Hsa-loops and **e** GST-SLBR_UB10712^Hsa-loops which contain the CD, EF, and FG loops of SLBR_Hsa. **a–e** data points represent the mean value and bars represent the standard deviation. **f** Quantitation of bound glycans at a concentration of 2 μg/ml to parent and chimeric SLBRs. Individual datapoints are shown in black dots with the bar height and the thin black bars representing the mean and standard deviation. The y axis is the absorbance at 450 nm of each sugar normalized to the absorbance of sTa to each SLBR. Root mean square deviation (rmsd) values were calculated from the normalized average signal of each glycan to compare the similarity of binding profiles between SLBRs. This identifies that both GST-SLBR_SK678^Hsa-loops and GST-SLBR_UB10712^Hsa-loops bind to sTa more strongly than the parent SLBRs. In addition, the chimeric SLBRs now have a preference for glycans more similar to SLBR_Hsa. Specifically, wild-type SLBR_SK678 and _UB10712 bind most strongly to 6S-sLe^X/3'sLn > sLe^X. In contrast, SLBR_Hsa and SLBR_UB10712^Hsa-loops bind sTa > sLe^C > 3'sLn > sLe^X > 6S-sLe^X while SLBR_SK678^Hsa-loops bound sTa > sLe^C/3'sLn > sLe^X/6S-sLe^X. **a-f** Measurements were performed using 500 nM of immobilized GST-SLBR and the indicated concentrations of each ligand (*n* = 3 independent experiments performed on protein from a single preparation). Source data are provided as a Source Data file.

(Supplementary Fig. 17a, b). We engineered SLBR_GspB-SK150 mini-chimeras that swapped single amino acids at these positions with the equivalent residues from SLBR_SK150. We then measured binding to sTa, 3'sLn, and sLe^C (Supplementary Fig. 17c–f). The SLBR_GspB^L442Y/Y443N mini-chimera had increased binding to 3'sLn and sLe^C and was overall more similar in selectivity to SLBR_SK150 than to SLBR_GspB (compare Supplementary Fig. 17c and Supplementary Fig. 1); however, the converse SLBR_SK150^Y300L/N301Y mini-chimera still exhibited reduced binding (Supplementary Fig. 17d) and a size exclusion profile that suggested the presence of breakdown products, indicating that misfolding likely contributes to loss of binding for this variant (Supplementary Fig. 16d). The incomplete success of the mini-chimeras suggests complex origins for the inability to change selectivity in SLBR_GspB and SLBR_SK150 via mutagenesis.

In summary, the SLBRs from the two branches of the evolutionary tree respond differently to chimeragenesis. The parent SLBR_GspB and SLBR_SK150 cannot easily undergo alteration of their binding spectrum and tend to exhibit lower stability

(Supplementary Fig. 16a–d) and loss of function (Supplementary Fig. 17c–f). In contrast, SLBR_Hsa, SLBR_SK678, and SLBR_UB10712 readily tolerate changes in binding spectrum via chimeragenesis to allow strong binding of alternative ligands (Supplementary Table 4).

**Identification of selectivity-dictating residues.** The identification of the CD, EF, and FG loops as the regions that are of largest natural sequence variation (Supplementary Fig. 4) and as regions that may control glycan selectivity (Fig. 6 and Supplementary Fig. 15) could suggest that these evolved to allow for binding to different host receptors. Natural evolutionary changes in SLBR sequence might involve point mutations rather than substitutions of entire loops. We therefore wanted to test whether point mutations of the loops of SLBR_Hsa, SLBR_SK678, and SLBR_UB10712 could change the selectivity. In SLBR_Hsa, SLBR_SK678, and SLBR_UB10712, we substituted residues at positions equivalent to SLBR_Hsa^E286 of the CD loop and SLBR_Hsa^D356 of the FG loop

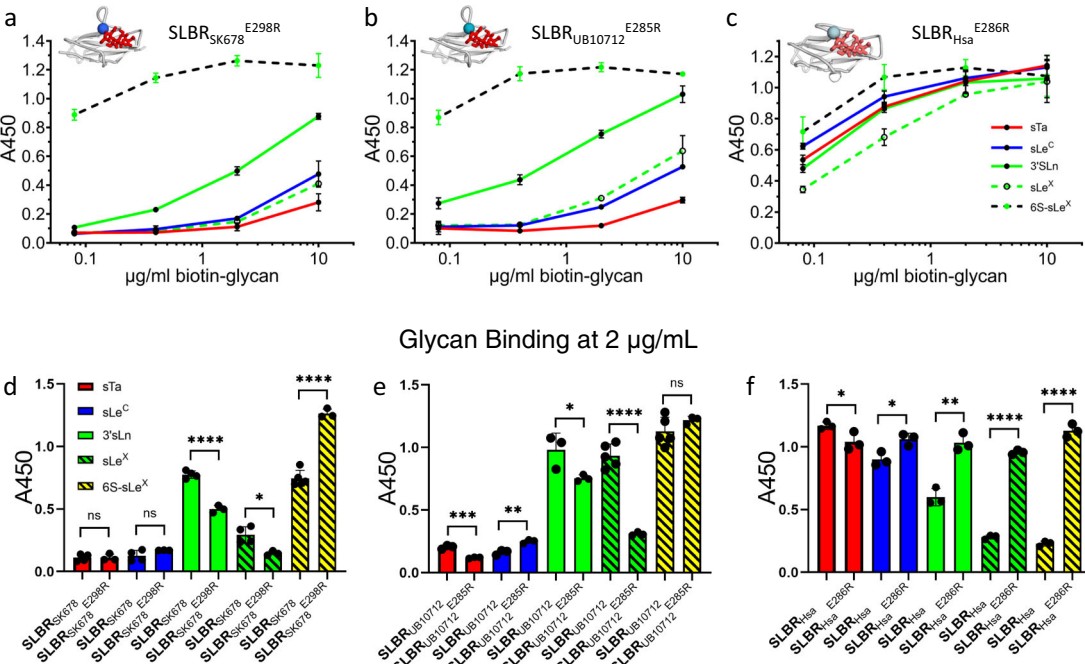

**Fig. 7 Binding selectivity of CD loop variants in SLBR$_{SK678}$, SLBR$_{UB10713}$ and SLBR$_{Hsa}$.** Dose response curves of biotin-glycan binding to immobilized variant GST-SLBRs (500 nM). **a** GST-SLBR$_{SK678}$E298R, **b** GST-SLBR$_{UB10712}$E285R, **c** GST-SLBR$_{Hsa}$E2866. The respective SLBRs are shown in gray cartoon in the top left corner of each panel with the site of mutation represented as a colored sphere. sTa, shown in red sticks, was placed over the binding site by superimposing sTa bound-SLBR$_{Hsa}$. Measurements were performed using 500 nM of immobilized GST-SLBR and the indicated concentrations of each ligand are shown as the mean ± SD. **d–f** Binding of each sugar at 2 µg/mL to each mutant was statistically compared to binding of the same sugar to the SLBR$^{WT}$ with the values presented in **a–c** using a two-tailed parametric $t$ test. Black circles represent individual data points and bars represent the mean ± SD ($n = 3$ independent experiments performed on protein from a single preparation). Statistical significance is indicated by: ns, $p > 0.05$; *, $p < 0.05$; **, $p < 0.01$; ***, $p < 0.001$; ****, $p < 0.0001$. In panel **d**, the $p$ values from left to right are 0.94, 0.15, <0.0001, 0.013, and <0.0001. In panel **e**, the $p$ values from left to right are 0.0007, 0.0034, 0.046, <0.0001, and 0.24. In panel **f**, the $p$ values from left to right are 0.047, 0.019, 0.0016, <0.0001, and <0.0001. Source data are provided as a Source Data file.

because our structures show that these residues closely approach the variable region of the ligands (Supplementary Fig. 13). We then measured relative binding to five physiologically-relevant ligands via ELISA (Fig. 7a–c and Supplementary Table 4).

In the CD loop (i.e., SLBR$_{Hsa}$E286), our crystallographic analysis suggested that ionic repulsion from the negatively-charged side chain excludes the negative charge of a sulfated ligand. We therefore substituted a positive charge at this location in SLBR$_{UB10712}$, SLBR$_{SK678}$, and SLBR$_{Hsa}$. All three of these variants exhibited a substantial increase in binding for 6S-sLe$^X$ (Fig. 7d–f and Supplementary Table 4). SLBR$_{Hsa}$E286R retained binding to non-sulfated ligands and this variant became quite promiscuous for the ligands tested by ELISA (Supplementary Fig. 15c). To better evaluate the binding spectrum of SLBR$_{UB10712}$ and SLBR$_{SK678}$, we assessed >500 glycans via array analysis as compared to a GST control (Supplementary Fig. 18 and Supplementary Data 1). These studies indicate that the engineered SLBRs are selective for two closely-related glycans: 6S-sLe$^X$ and 6S-3'sialyllactosamine (6S-3'sLn, Neu5Acα2-3Galβ1-4GlcNAc6Sβ, Fig. 1g) which lacks the fucose.

We then evaluated selectivity conferred by the FG loop where crystallographic analysis would suggest that the β-branching of SLBR$_{Hsa}$D356 excludes C3 fucosylation, while the larger, unbranched Gln of SLBR$_{UB10712}$ and SLBR$_{SK678}$ can bind fucosylated ligands. We therefore substituted Asp for Gln in SLBR$_{UB10712}$ and SLBR$_{SK678}$ and conversely substituted Gln for Asp in SLBR$_{Hsa}$. As assessed by ELISA, the SLBR$_{SK678}$Q354D and SLBR$_{UB10712}$Q367D variants lost binding to fucosylated ligands (Fig. 8a, b and Supplementary Fig. 19a, b). As a result, SLBR$_{UB10712}$Q354D became more selective for 3'sLn while the

SLBR$_{SK678}$Q367D exhibited low binding to all tested ligands. As assessed by size exclusion chromatography, the SLBR$_{SK678}$Q367D variant was properly folded such that loss of binding did not result from a folding defect (Supplementary Fig. 19c). The observed loss of binding to the fucose-containing sLe$^X$ and 6S-sLe$^X$ by these FG loop variants is consistent with the structural analysis and chimeragenesis showing that the FG loop is particularly important for accommodation of α1,3-fucosylation (Fig. 6 and Supplementary Fig. 15a, b). The converse SLBR$_{Hsa}$D356R, and SLBR$_{Hsa}$D356Q remained broadly-selective but with increased binding to the α1,3-fucosylated sLe$^X$ and 6S-sLe$^X$ as compared to parent SLBR$_{Hsa}$ (Fig. 8c, d and Supplementary Fig. 19d).

Taken together, point mutations in the broadly-selective SLBRs can alter the identity of the preferred ligand, and can bind robustly to the newly-preferred ligand. The EC$_{50}$ values suggest that the binding is strong enough to make physiologically-relevant adhesive interactions to host receptors. A possible evolutionary rationale for facile alteration in sialoglycan binding spectrum is that this allows a bacterium to adapt to changes in host sialoglycan display.

**Selectivity variants alter the preferred host receptor.** To test whether changes in SLBR binding to synthetic glycans had corresponding effects in the interactions of the SLBRs with human ligands, we examined the binding of parent and variant SLBRs to human salivary and plasma glycoproteins using far western analysis. We focused on the chimeras and variants that had narrower selectivity, where changes in binding would be most

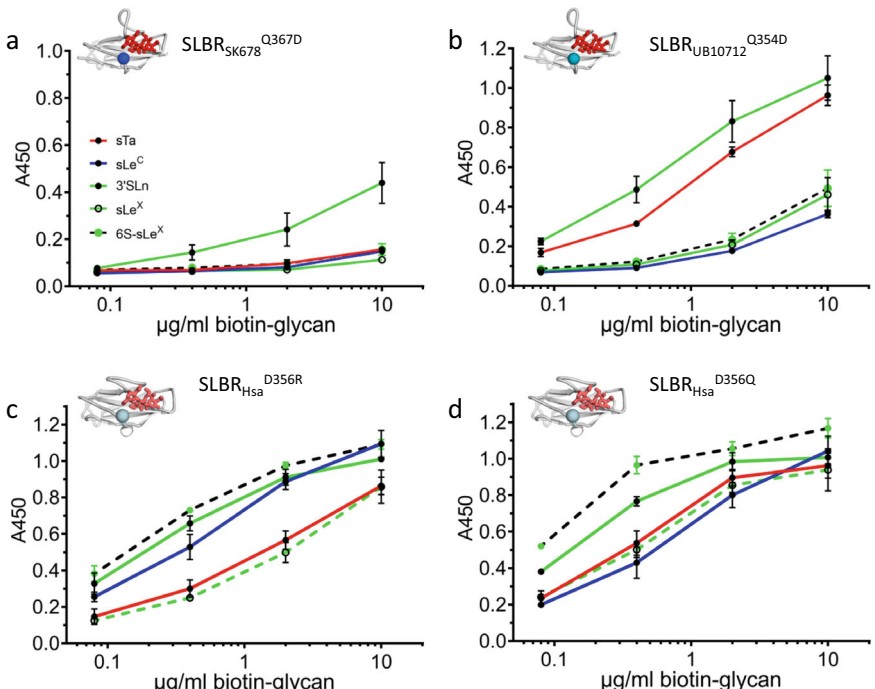

**Fig. 8 Binding selectivity of FG loop variants in SLBR$_{SK678}$, SLBR$_{UB10713}$, and SLBR$_{Hsa}$.** Dose response curves of biotin-glycan binding to immobilized variant SLBRs (500 nM). Both **a** the GST-SLBR$_{SK678}$$^{Q367D}$ variant and **b** the GST-SLBR$_{UB10712}$$^{Q345D}$ variant have substantially reduced binding to the fucosylated ligands sLe$^X$ and 6S-sLe$^X$. In SLBR$_{Hsa}$, charge reversal or neutralization at this same position was assessed in **c** GST-SLBR$_{Hsa}$$^{D356R}$ and **d** GST-SLBR$_{Hsa}$$^{D356Q}$. Both variants had increased binding to 6S-sLe$^X$, 3'sLn, and sLe$^X$ and decreased binding to sTa, albeit to somewhat different extents. Measurements were performed using 500 nM of immobilized GST-SLBR and the indicated concentrations of each ligand are shown as the mean ± SD ($n = 3$ independent experiments with a single protein preparation). Statistical comparisons of ligand affinity between FG mutants and parent SLBR can be found in Supplementary Fig. 18. Source data are provided as a Source Data file.

evident. We first identified the glycoprotein targets of parent and variant SLBRs in submandibular sublingual (SMSL) ductal saliva from four donors. The three parent SLBRs recognized a band consistent with the mobility of MUC7 in all four samples (Fig. 9a), but the levels of binding differed between samples. The bands were excised from a gel and analyzed with LC/MS (Supplementary Data 2, 3). In addition, the SLBRs bound MUC7 glycoforms of different apparent mass ranges, likely reflecting differences in the number, size and composition of *O*-glycan structures[20,21]. SLBR$_{SK678}$ and SLBR$_{UB10712}$ detected glycoforms of ~160 kDa, whereas SLBR$_{Hsa}$ bound more readily to 140–150 kDa glycoforms (Fig. 9a). SLBR$_{UB10712}$ recognized the band consistent with MUC7 in all four samples nearly equally, whereas SLBR$_{SK678}$ detected this band from donor 3 > donors 1 and 4 > donor 2, and SLBR$_{Hsa}$ detected this band from donor 3 > donors 2 and 4 > donor 1. The recognition pattern of the SLBR$_{SK678}$$^{Hsa-loops}$ and SLBR$_{UB10712}$$^{Hsa-loops}$ chimeras resembled that of SLBR$_{Hsa}$ rather than that of the parent SLBR$_{SK678}$ and SLBR$_{UB10712}$. These loop exchanges altered both the apparent mass recognized and the avidity of the binding. In contrast, the 6S-sialoglycan-selective point mutants showed preferential binding to the uppermost mass range of MUC7 in samples from donors 1 and 4, and a near loss of binding to samples from donors 2 and 3.

We next determined whether the recognition patterns correlated with the presence of sTa versus 3'sLn (for the loop chimeras) or with the presence of 6-*O*-sulfo structures (for the single residue substitutions), decorating larger physiological glycans. To do this, we used affinity capture and mass spectrometry to characterize the *O*-glycan composition of the four MUC7 samples (Fig. 9b and Supplementary Figs. 20 and 21). The *O*-glycan profiles were similar to those seen in two earlier

reports[20,21], in that dozens of different structures were evident in each sample. The most abundant structures were mono- or di-sialylated Core 2 glycans. There were relatively minor amounts of sTa and there were differences in the assortment of other minor structures. The glycans from the four donors differed in the extent of sialylation and fucosylation (Supplementary Figs. 20 and 21), the presence or absence of sulfated structures (Fig. 9b), and the relative abundance of each species. The *O*-glycan profiles are consistent with the ELISA measurements to purified glycans (Fig. 6c–e). Specifically, SLBR$_{Hsa}$, SLBR$_{SK678}$$^{Hsa-loops}$, and SLBR$_{UB10712}$$^{Hsa-loops}$ preferred sTa in the ELISA assays with purified glycans (Fig. 6) and bound Core 2 structures that contain Neu5Ac on the sTa-like Core 1 branch in salivary MUC7 (Fig. 9b). In addition, SLBR$_{SK678}$ and SLBR$_{UB10712}$ bound to 3'sLn and 6S-sLe$^X$ in ELISA assays (Fig. 6a, b) and bound to structures that have Neu5Ac on the 3'sLn branch in MUC7 (Fig. 9b). Finally, the SLBR$_{SK678}$$^{E298R}$ and SLBR$_{UB10712}$$^{E285R}$ both strongly preferred 6-*O*-sulfated species over other ligands (Fig. 7). The presence of a 6S-3'sLn moiety in the samples from donors 1 and 4 (the 2-2-0-2-1 structure) suggests that these variants recognize MUC7 modified with relatively minor amounts of 6S-3'sLn, potentially reflecting high-affinity binding.

SLBRs may also interact with glycoproteins in the bloodstream, and the binding spectrum may have consequences for pathogenicity. We therefore next evaluated binding to human plasma proteins by far western analysis. Consistent with our prior studies, parent SLBR$_{Hsa}$ preferentially bound proteoglycan 4 (460 kD) from human plasma, while SLBR$_{UB10712}$ bound GPIbα (150 kD). Of note, proteoglycan 4 is a major carrier of sTa in plasma, whereas GPIbα has predominantly di-sialylated Core 2 structures. These SLBRs also bound different glycoforms of the C1-esterase inhibitor (100–120 kDa)[8] (Fig. 9c). The

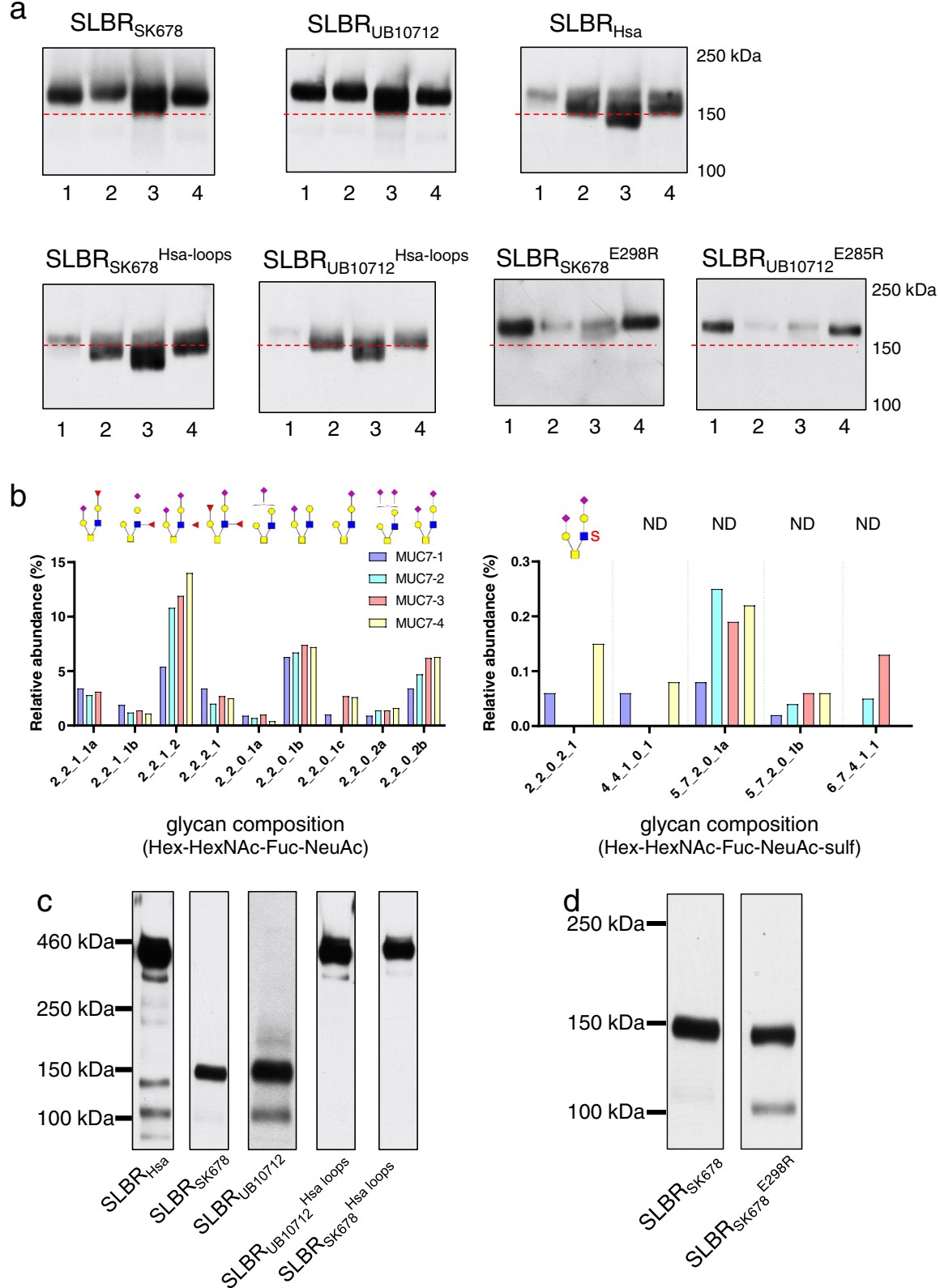

chimeric SLBR$_{UB10712}$$^{Hsa-loops}$ and SLBR$_{SK678}$$^{Hsa-loops}$ chimeras now recognized proteoglycan 4 rather than the preferred receptors for parent SLBR$_{SK678}$ and SLBR$_{UB10712}$ (Fig. 9c). We also found that the SLBR$_{SK678}$$^{E298R}$ variant bound both GPIbα, a receptor associated with infective endocarditis, and the C1-esterase inhibitor (Fig. 9d). Thus, the preferred plasma ligands for the SLBRs appears to be largely determined by the

loop residues, as was the case for the recognition of MUC7 glycoforms.

## Discussion

Bacterial attachment to host structures is critical for commensalism and is the first committed step in many types of infection. SLBRs can mediate streptococcal binding to a variety of host

**Fig. 9 MUC7 O-glycans and SLBR recognition of glycoproteins in human saliva and plasma. a** Representative far-western blot of the SMSL saliva samples with parent and variant GST-tagged SLBRs ($n = 2$). The MUC7 glycoforms range from 120 to 160 kDa. Saliva samples (1 μl) were run on the same gel and transferred to the same nitrocellulose membrane. The membrane was subsequently cut in order to separately probe with parent versus variant SLBRs (15 nM). The dashed red line indicates the 150kD molecular weight marker. Uncropped blots source data are provided as a Source Data. **b** The major non-sulfated (left) and sulfated (right) O-linked glycans from MUC7 in four samples of submandibular sublingual (SMSL) saliva. The $x$-axis represents glycan compositions Hex-HexNAc-Fuc-Neu5Ac and Hex-HexNAc-Fuc-Neu5Ac-Sulf for the upper left and right panel, respectively. Lower case letters a, b, and c indicate different isomer structures with the same monosaccharide compositions. Putative structures are shown above the graphs (ND not determined). **c** Representative far-western blot of human plasma with parent and chimeric GST-tagged SLBRs ($n = 2$). As previously identified by affinity capture and mass spectrometry[8], the 460 kD band is proteoglycan 4, the 150 kD band is GP1bα, and the 100 kD band is C1-esterase inhibitor. **d** Representative far-western blot of human plasma with parent SLBR$_{SK678}$ and the SLBR$_{SK678}$$^{E298R}$ point mutant ($n = 2$). Source data are provided as a Source Data file.

glycoproteins[5,7–10,14,22,24,25,29,30], and binding to sTa correlates with pathogenesis in an animal model of endovascular infection[22]. But it has not previously been clear how the SLBRs distinguish between the many protein-attached glycans displayed by host. Here, we evaluated how five SLBRs select between sialoglycan receptors. The common element of these glycans, Neu5Acα2-3Gal, interacts with SLBRs via the ΦTRX motif[5,13,16,31] and the EF loop (Fig. 4 and Supplementary Fig. 9)[16,23]. The CD and FG loops select for the underlying reducing end (Figs. 6, 8, and Supplementary Figs. 15 and 17), which varies in the identities of its individual sugars, the linkage between the sugars, and the elaborations on the sugars (Fig. 1). This suggests roles for distinct regions of the SLBR structure in glycan selection (Fig. 10) The substantial sequence and structural variability in the CD, EF, and FG loops as compared to the core fold of the SLBR (Supplementary Figs. 3 and 4) suggests that these regions can tolerate more substitutions while avoiding the liability of misfolding. Indeed, modification of these regions via chimeragenesis or mutation allowed some of the SLBRs to bind different glycoforms of MUC7 or interact with different preferred sialoglycans (Figs. 6, 7, 8, and Supplementary Figs. 14, 15, 17 and 20) and different host plasma proteins (Fig. 9).

Although not previously noted for bacterial SLBRs, the use of loops to control selectivity has been observed in other sialoglycan-binding systems. For example, mammalian Siglec proteins are organized around a V-set Ig-fold but are not detectably related in sequence to the SLBRs[13,23,32,33]. The GG' and CC' loops are adjacent to the sialoglycan binding site and are variable in structure. In Siglec-7, the CC' loop[34] controls sialoglycan selectivity. In Siglec-8, alteration of this same loop allows the binding of 6'S sialoglycans[35]. Thus, changes in loop structure may therefore be a common way to evolve changes in ligand binding selectivity.

The use of loops to control selectivity appears to be a robust way to accommodate a broad range of complex glycans. Indeed, the glycans recognized by SLBRs differ in both the identity of the individual glycans as well as in the linkages between the individual carbohydrates. When bound to these SLBRs (see Supplementary Fig. 13b), glycans with different linkages differ in the overall shape as well as in the pattern of hydrogen-bonding donors and acceptors. However, the glycosidic linkage itself does not differ in position with respect to the SLBR binding pocket. Thus, these SLBRs distinguish between glycans with different linkages by changing the steric and electrostatic properties of the region of the pocket that follows the linkage, namely the CD and FG loops. While we focused on SLBRs that recognize tri- and tetrasaccharides, SLBRs can recognize sialoglycans with as few as three and possibly more than six monosaccharide units[5,8,19,23]. For example, SLBR$_{SrpA}$ may biologically recognize a hexasaccharide[8] but can bind to partial ligands with lower affinity[5,16,23]. SLBRs that recognize larger sialoglycans appear to contain a modular binding site similar to those studied here, albeit with larger binding pockets and with more independent recognition regions. In the oral cavity,

this may assist in colonization through interaction with salivary MUC7, which exhibits heterogeneity of its sialoglycan modifications both within and between human hosts (Fig. 9a, b and Supplementary Figs. 20 and 21). Here, sialoglycans are attached to MUC7 and the SLBR binding pocket can bind glycan receptors that are linked to host proteins. The linkage to the receptor protein could affect binding and could involve additional contacts to the SLBR[18].

In this context, mutation of these loops may be advantageous to the bacterium because it allows facile switching of host receptors. While we do not know how the sequences of the SLBRs actually change during evolution, streptococci compete with numerous other species in the oral cavity[36]. As many of these strains contain SLBRs, genetic recombination is likely, which can allow a bacterium to incorporate or modify a SLBR. The ready toleration of mutations in the loops may allow these regions to disproportionately change their sequences. Some of these changes may enable the bacterium to bind a different sialoglycan structure (Figs. 7–9 and Supplementary Figs. 15, 18 and 19). Within a single human host, this could allow colonization of a region of the oral cavity that displays different glycans, could promote binding to different salivary components, or could allow binding to other oral bacteria that are sialylated. This mutability could also permit improved binding to different individuals in the population or allow the colonization of a preferred host, as animals and humans may differ in their glycosylation[37]. This mechanism mirrors that of polyomavirus and rotavirus, where single amino acid substitution or a very small number of point mutations can change the identity of preferred host sialoglycan receptors[38,39].

In some of our point mutants, the improvement in affinity and selectivity to alternative ligands exceeds those reported for dedicated engineering studies of glycan-binding lectins[40–48]. In those past reports, the maximum enhancement in binding to a non-native glycan is ~20-fold[40–45] and selectivity was often achieved via a decrease in affinity to non-desired ligands in a promiscuous starting lectin[46–48]. Development to increase the affinity and narrow the selectivity even further could allow the SLBRs to be used as probes to assess glycan identity and abundance. Key aspects of a probe include the ability to detect glycans in cells and in patient samples. The cellular interaction was shown in recent studies that evaluated the binding SLBRs to engineered HEK293 cell lines with altered glycosylation[49], while the ability to recognize glycans in saliva and plasma suggests that these will be useful in other samples (Fig. 9).

Collectively, our findings give a description for how SLBRs recognize ligands. The conserved sialic acid-recognition motif governs general specificity while sequence diversity in surrounding loop regions allows the SLBR to select between related sialoglycans (Fig. 10). This binding site architecture may be optimized for facile selectivity changes in related SLBRs. This may further explain how bacterial adhesive proteins have evolved to adapt to host receptors. Finally, this work suggests a route for engineering these SLBRs to use as probes to detect specific

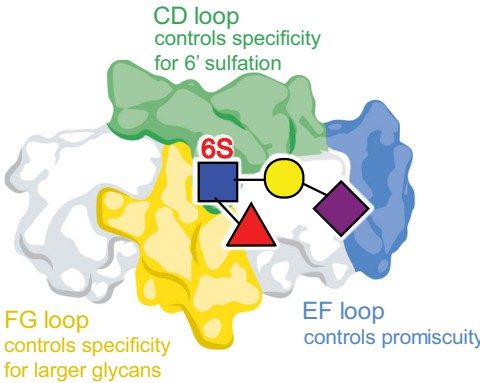

**Fig. 10 Model for how SLBRs control sialoglycan selectivity.** The glycan-binding pocket of SLBRs is organized above a ΦTRX sequence motif on the F-strand of the V-set Ig fold that interacts with sialic acid. Three variable loops surround this sialoglycan binding pocket and affect selectivity. In the broadly-selective SLBRs, flexibility of the EF loop correlates with breadth of selectivity. The CD loop controls specificity for 6-O-sulfated glycans, and the FG loop may control whether the SLBR prefers trisaccharides versus larger glycans. Tetrasaccharides containing α1,3-fucosylation were tested here, but past studies of SLBR$_{SrpA}$ identify that a small FG loop correlates with the ability to accommodate larger glycans[23].

glycosylation, which is a focus of ongoing work. A library of SLBR-based binding proteins could be used for glycome mapping or as diagnostic or therapeutic tools for disease states with aberrant glycosylation.

## Methods

**Sequence analysis.** SLBR sequences were aligned using the MUSCLE[50] subroutine in Geneious Pro 11.1.4[51]. The JTT-G evolution model was selected using the ProtTest server[52], and the phylogenetic tree was built using the MrBayes[53] subroutine.

**Cloning, expression, and purification for crystallization.** DNA encoding all SLBRs except SLBR$_{Hsa}$ were cloned into the pBG101 vector (Vanderbilt University), which encodes an N-terminal His$_6$-GST tag cleavable with 3C protease. SLBR$_{Hsa}$ was cloned into the pSV278 vector (Vanderbilt University), which encodes a thrombin-cleavable His$_6$-maltose binding protein (MBP) tag. Proteins were expressed in *E. coli* BL21 (*DE3*) with 50 µg/ml kanamycin at 37 °C. Expression was induced with 0.5–1 mM IPTG at 24 °C for 3–7 h. Cells were harvested by centrifugation at 5000 × *g* for 15 min and stored at −20 °C before purification.

Cells were resuspended in 20–50 mM Tris-HCl, pH 7.5, 150–200 mM NaCl, 1 mM EDTA, 1 mM PMSF, 2 µg/ml Leupeptin, 2 µg/ml Pepstatin then disrupted by sonication. Lysate was clarified by centrifugation at 38,500 × *g* for 35–60 min. Tagged fusion proteins were purified using a Glutathione Sepharose 4B column eluted with 30 mM GSH in 50 mM Tris-HCl, pH 8.0, a Ni$^{2+}$ affinity chromatography eluted with 20 mM Tris-HCl, 150 mM NaCl, 250 mM imidazole, pH 7.6, or a MBP-Trap column eluted in 10 mM maltose. Affinity tags were cleaved with 1 U of protease per mg of protein overnight at 4 °C. Protein was separated from the cleaved affinity tag by passing over the relevant affinity column. Protein aggregates were removed using either a Superose-12 column in 50 mM Tris-HCl pH 7.6 and 150 mM NaCl or a Superdex 200 increase 10/30 GL column equilibrated in 20 mM Tris-HCl pH 7.6 or in 20 mM Tris-HCl pH 7.5 and 200 mM NaCl.

We note that the *S. gordonii* strain UB10712 was recently re-typed. Previous literature refers to this strain as *S. mitis* strain NCTC10712.

**Structure determination.** Crystallizations were performed at room temperature (~23 °C) using the conditions in Supplementary Table 5. The SLBR$_{GspB}$-sTa structure used crystals where the ligand was introduced by cocrystallization, and the SLBR$_{Hsa}$-ligand structures used crystals where the ligand was introduced by soaking. Data collection and refinement statistics are listed in Supplementary Tables 1, 2, and 3. Structures were determined by molecular replacement using the Phaser[54] subroutine of Phenix 1.18.2[55] using the starting models listed in Supplementary Table 5.

All models were improved with iterative rounds of model building in Coot 0.9[56] and refinement in Phenix 1.18.2[55]. Riding hydrogens were included at resolutions better than 1.4 Å. For sialoglycan-bound SLBR$_{Hsa}$, the crystals were isomorphous

with unliganded crystals and R$_{free}$ reflections were selected as identical. Ligand occupancies were held at 1.0 during refinement. Representative electron density maps for the ligand-bound structures can be found Fig. 4, while representative density for the unliganded structures can be found in Supplementary Fig. 22.

**Sialoglycan binding.** DNA encoding wild-type and variant SLBRs were cloned into pGEX-3X. Individual GST-SLBR fusions were expressed and purified using glutathione-sepharose, and the binding of biotinylated glycans to immobilized GST-SLBRs was performed as described previously[5]. Anti-GST antibody was used at a 1:500 dilution and was from Invitrogen (A5800). Peroxidase-conjugated goat anti-rabbit IgG was used at a 1:5,000 dilution and was from Sigma (A0545). The number of replicates of each data point are in each figure legend. Replicates are independent replicates from separately-prepared samples.

**Far western and lectin blotting of human proteins.** Far-western blotting of human saliva and plasma proteins using the indicated GST-SLBRs (15 nM) as probes was performed as described[5,8]. Plasma was purchased from Innovative Research (Novi, MI). De-identified samples of SMSL saliva were provided by S. Fisher (UCSF), and were collected through a protocol approved by the UCSF Institutional Review Board. Donors confirmed that their samples may be used for other research purposes. Because these specimens were de-identified prior to gifting, our use of this material was exempt from approval by the UCSF Institutional Review Board and was not classified as human subject research. Anti-GST antibody was from Invitrogen (A5800) and peroxidase-conjugated goat anti-rabbit IgG was from Sigma (A0545). Uncropped gels source data are provided as Source Data file.

**MUC7 affinity capture and O-glycan profiling.** A combination of GST-SLBR$_{Hsa}$ and GST-SLBR$_{UB10712}$ immobilized on magnetic glutathione beads was used to capture the total sialylated MUC7 from 300 µl of SMSL saliva. The resin-bound GST-SLBRs and affinity-captured MUC7 were co-eluted into LDS sample buffer (Invitrogen) supplemented with dithiothreitol (100 mM final concentration), separated by electrophoresis in 4–12% polyacrylamide gradient gels, and then stained with SimplyBlue SafeStain (Invitrogen). The captured proteins, which ranged from 120–160 kDa, were excised from the gel. A portion of the sample was submitted for protein identification by nanoflow LC-MS/MS of tryptic digests (MSBioworks), which confirmed MUC7 as the major component. A second portion of the excised gel slices was minced, treated by four cycles of rinsing with 100 mM ammonium bicarbonate and dehydration in 100% acetonitrile, and then dried to completion in a vacuum evaporator. The gel pieces were immersed in a mixture of 100 mM NaOH and 1 M NaBH$_4$ and incubated at 45 °C for 18 h to release the O-glycans. The supernatant was collected and placed on ice, and the remaining gel pieces were washed with water and sonicated for 30 min to extract the remaining O-glycans. The initial and secondary extracts were combined and acidified to pH 4-6 by drop-wise addition of 10% acetic acid. The O-glycan samples were then enriched using porous graphitized carbon cartridges (Agilent, Santa Clara, CA) and dried prior to analysis by mass spectrometry. Glycan samples were analyzed on an Agilent 6520 Accurate Mass Q-TOF LC/MS equipped with a porous graphitic carbon microfluidic chip. A binary gradient consisting of (A) 0.1% formic acid in 3% acetonitrile, and (B) 1% formic acid in 89% acetonitrile was used to separate the glycans at a flow rate of 0.3 µl/min. Data were processed with Agilent MassHunter B.07 software, using the Find by Molecular Feature algorithm with an in-house library of O-glycan masses and chemical formulae to identify and quantitate the O-glycan signals.

**In silico structure predictions and MD analyses.** The model of SLRB$_{SK678}$$^{Hsa-loops}$ was calculated using MOE. For MD of SLBR$_{Hsa}$, SLBR$_{GspB}$, SLBR$_{SK678}$, and SLBR$_{SK678}$$^{Hsa-loops}$ each set of PDB coordinates was solvated in a 10 Å octahedral box of TIP3P[57] water. The Amber16 ff14SB[58] force field was used for the protein. In the first step of the MD simulation, the backbone and side chains of the protein were restrained using 500 kcal mol$^{-1}$ Å$^{-2}$ harmonic potentials while the system was energy minimized for 500 steps of steepest descent[59] and the conjugate gradient method[60]. Restraints were removed and 1000 steps of steepest descent minimization were performed followed by 1500 steps of conjugate gradient. The system was then subjected to MD at 300 K with the backbone and side chains restrained using 10 kcal mol$^{-1}$ Å$^{-2}$ harmonic potentials for 1000 steps. Bonds were constrained using SHAKE[61]. MD (200 ns) was performed at 300 K in the NPT ensemble and a 2-fs time step. Probability distribution analyses and RMSF calculations were performed on 200 ns of 3 independent runs. Analyses were performed using the cpptraj and pytraj[62] modules of AMBER16. The last snapshot from 20-ns trajectory was used for mapping the interaction between the glycans and SLRB$_{SK678}$ or SLBR$_{SK678}$$^{Hsa-loops}$.

**Reporting summary.** Further information on research design is available in the Nature Research Reporting Summary linked to this article.

## Data availability

Source data are provided as a Source Data file. Atomic coordinates and structure factors have been deposited into the RCSB Protein Data Bank at www.rcsb.org under the accession codes

6EFA, 6EFB, 6EFC, 6EFD, 6EFF, 6EFI, 6EF7, 6EF9, 6X3Q, 6X3K, 7KMJ. Previously published structures shown are available via the accession codes 5IJ3, and 6VT2. Previously published structures used for molecular replacement are available via the accession codes 5EQ2, and 3QC5.

Raw data have been deposited into SBGrid (data.sbgrid.org) with the accession codes 328, 329, 507, 508, 509, 510, 601, 604, 787, 788, 812, and 813. Glycomics data were deposited in MassIVE (https://massive.ucsd.edu/) with the data identifier MSV000088327.

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

## Acknowledgements
We thank S Bordenstein, L Loukachevitch, and P Singh for experimental and analytical assistance, and B Bachmann, R Woods, and O Grant for helpful discussions. This work was supported by the Department of Veterans Affairs, the National Institutes of Health (R01AI41513 and U01CA221244 to P.S., R01AI130684 to X.C., R01AI106987 to T.M.I./P.M.S., R03 DE029516 to B.A.B., and GM137458 to T.M.I.), and the American Heart Association (14GRNT20390021 to T.M.I.; 17SDG33660424 to B.A.B.). AH was supported by the Vanderbilt International Scholars Program, and MAC was supported by a National Science Foundation Individual pre-doctoral fellowship (DGE-1445197). K.M.M. was supported by National Institutes of Health Training grant GM007628. H.E.S. was supported by National Institutes of Health Training grants GM008320 and EY007135. Use of the Stanford Synchrotron Radiation Lightsource, SLAC National Accelerator Laboratory, is supported by the U.S. Department of Energy, Office of Science, Office of Basic Energy Sciences under Contract No. DE-AC02-76SF00515. The SSRL Structural Molecular Biology Program is supported by the DOE Office of Biological and Environmental Research, and by the National Institutes of Health, National Institute of General Medical Sciences (including P41GM103393). The Advanced Photon Source, a User Facility operated for the U.S. DOE Office of Science, was supported under Contract DE-AC02-06CH11357. LS-CAT Sector 21 is supported by the Michigan Economic Development Corporation and the Michigan Technology Tri-Corridor (085P1000817). The Protein-Glycan Interaction Resource of the CFG is supported by the National Institutes of Health, National Institute of General Medical Sciences (R24 GM098791) and the National Center for Functional Glycomics (NCFG) at Beth Israel Deaconess Medical Center, Harvard Medical School (P41 GM103694).

## Author contributions
T.M.I. and B.A.B. designed research, B.A.B., I.Y., H.E.S., K.L., K.S., A.H., R.A., K.M.M. and Z.W. performed research, B.A.B., H.E.S., M.A.C., K.L., K.S., I.Y., K.P.F. and T.M.I. analyzed data, H.Y. and X.C. contributed reagents/analytical tools. C.B.L., J.B., J.C.S., P.M.S. and T.M.I. conceived and guided the study. T.M.I. and B.A.B. wrote the manuscript with input from other authors. All authors approved the final version of the manuscript.

## Competing interests
T.M.I., P.M.S., and B.A.B. hold a provisional patent, PCT/US2021/036983 that covers mutation of SLBRs for use as binding probes. The patent includes mutants described here as well as mutations with different selectivities and the methods to create new mutants via mutation or chimeragenesis. Authors on the patent are affiliated with Vanderbilt University, the Regents of the University of California, and the United States as represented by the Department of Veterans Affairs. The remaining authors declare no competing interests.
