## [Peer Review File · Nature Communications]

REVIEWER COMMENTS

Reviewer #1 (Remarks to the Author):

Review for Bensing et al. Submitted to Nature Communications

As a bacterial geneticist with an interest in streptococcal carbohydrate binding proteins I am not qualified to comment on the appropriateness of the structural, or some of the biochemical, methods and much of the data analysis. I limit my review to the biological relevance, impact on the field, and the ability of a highly interested individual with different expertise to comprehend the data.

Siglec-like domain containing proteins are important virulence factors in many streptococci that colonize the oral cavity and can cause infective endocarditis. Their role in the oral cavity is less defined, but they can bind sialylated structures on mucins and are likely critical in that environment too. These Siglec-like domain containing proteins all bind sialic acid but can have different binding specificities. Despite structural analysis of some Siglec-like domains we are currently unable to determine binding specificity without a glycan array and even then, the mechanism which defines binding specificity was unclear. This manuscript transforms our understanding of the binding mechanism. It has been proposed that three loops in the Siglec-like domains that contribute to sialic acid binding, but this is the first time that clear evidence has been presented. An extensive series of structural and biochemical studies puts together a compelling story that elucidates the roles for these three loops in the binding specificity of the different Siglec-like domains. The Authors demonstrate that exchange of loops or specific point mutations can alter binding specificity. Furthermore, they examine how these differences altered binding to salivary and serum proteins. Although the diversity of Siglec-like domains in individual species is poorly understood these experiments suggest that strains colonizing different individuals may be selected for by the glycan profile on salivary proteins and that some individuals may be more prone to developing infective endocarditis. The studies included in this manuscript not only shed light on the binding mechanism of these important proteins, but also open the door to development of a panel of specific binding Siglec-like binding proteins that could be used as a tool to determine the presence of specific carbohydrate structures. Glycobiology is currently a difficult field of study due to the lack of tools, but it is a critical area that will be more widely studied in the future.

I believe this manuscript will appeal to a broad range of scientists including those studying colonization and pathogenic mechanisms of streptococci, those studying adhesins and sialic acid binding proteins, glycobiologists, and structural biologists that might well find parallels in other systems.

In general, the manuscript is very well written, and a great deal of thought has gone into the organization of the figures to make a very complex set of analyses understandable by a broad

audience. To be honest, it is still pretty dense, but this is difficult to avoid while covering such a large scope. Some additional experiments are likely required and there are no doubt places where this Reviewer feels that clarity could be increased, but there is no doubt this was a massive undertaking that significantly moves the field forward.

Despite the strength of the manuscript there are some concerns

Data analysis and resulting interpretation and conclusions.

1) There is essentially no statistical analysis of any differences identified in the manuscript. Furthermore, in some cases it is not clear why $n=2$ and whether those were replicates in the same experiments or independent experiments. There would be more confidence in the data if three independent experiments were performed. This also complicates the Authors requests that we compare what appears to be across experiments to determine the effect of chimeras and point mutations. Maybe they were conducted at the same time but if not, it is critical to know that experiments were reproducible. The lack of a clear way to define what constitutes a difference is important and particularly so where I did not see the difference described by the Authors. Examples are given below:

Page 5: The Authors state “In contrast to the results observed with SLBR Hsa and its close homologs, substitution of the EF loop of SLBR SK150 into SLBR GspB had little impact”. However, I do not see much impact of the EF switch for Hsa binding to 3’sLn – in Fig. S9. Both chimeras and WT proteins bind this carbohydrate well. Also, the Author’s state “in all remaining chimeras, there was substantially decreased binding to both sTa and 3’sLn (Fig. S9C)”. Is this true? the WT binds poorly and so do the chimeras.

Fig. S9 There was a moderate decrease in SLBR UB10712 Hsa-FG-loop binding to sLeC does not appear to match the data. Furthermore, the legend includes several statements about small differences which it is difficult to determine whether they are significant.

The legend of figure 8 states “Both variants had increased binding to 6S-sLeX and 3’sLn and decreased binding to sTa and sLeX” is that actually true? I don’t see a reduction for the variants in binding to sLeX.

The legend of figure 7 states “When compared to wild-type (see Fig. 6A,B), the GST-SLBR SK678 E298R and GST-SLBR UB10712 E285R variants exhibit increased binding to 6S-sLeX, and reduced binding to 3’sLn and sLeX” Is this true? I don’t see much of a difference for SK678 binding to sLeX.

2) Much of the data is based on chimeras how is it known that substitution of loops doesn't prevent proper protein folding? I get they are loops, but can this be ruled out? This possibility is easy to eliminate in the case that some binding is increased but for some of the chimeras, such as many of the GspB-SK150 chimeras, no binding was observed. If this is a possibility it should be more clearly acknowledged, and conclusions tempered. Could proper folding be demonstrated by CD or other methodology?

3) Blots. It is hard to see the small difference in size of protein bound by the different Siglec-like domains in 9B. To support the claim they are different, it would seem important to run the samples on the same gel. Furthermore, there is no definitive evidence presented that these proteins are binding MUC7 or the plasma proteins referenced – cited work relies on a correlation of size to previous studies – can the Authors really exclude the possibility of other sialylated proteins running at these sizes in these donors? Can the Authors please confirm binding is to specific proteins if they wish to draw these conclusions. Or perhaps it doesn't matter, and it can just be hypothesized due to previously reported sizes. The Authors certainly show a difference in binding patterns when loops are switched.

Conclusions that are unclear and could be clarified

I think the conclusions/suggestions around evolution lacks some clarity. I understand space is limited, but I think this information needs to be clarified and more clearly presented. These bacteria are naturally transformable and hence likely undergo frequent horizontal gene transfer. They also exist in an environment where many other strains of oral streptococci that also contain Siglec-like domains are present making genetic recombination more likely. I think the fact that there are more changes in the loops observed as these regions can tolerate more differences should be included in the discussion. Some of these changes will alter the binding range and perhaps enable the bacterium to bind a different sialic acid structure, which could allow colonization of a different region of the oral cavity (I believe there is some evidence glycans are different in saliva at different sites in the oral cavity) or binding to different salivary components, or different individuals in the population. There might even be the opportunity to bind different sialylated bacteria (theoretically at least). It is not clear what the Authors mean by new anatomical location or in a new host – are you suggesting different animals or body sites – or the narrower ideas given above. It is also not clear what is meant by "it may also favor pure commensalism versus pathogenesis". The selective pressure for evolution of these species is in the mouth – I don't believe there is any selective pressure for or against bacteria binding to platelets or the heart valve. Perhaps that sentence can be clarified.

Representation of the field

In the introduction the Authors state that we do not know what dictates whether these bacteria can switch from commensal to pathogen, but do we even know that there are oral streptococci within

these species that are unable to become pathogens in humans? These studies are needed, and it is certainly likely that some species can cause disease and others cannot, but I don't of any studies that address this. Can more information be given or the sentence altered to more accurately reflect the understanding in the field?

Other concerns

In some of sections of the paper it reads to this Reviewer as if the bacteria are making conscious decisions. For example, around evolution – it reads that it was a decision on the part of the bacteria not random events that were maintained if they provided a selective advantage. Another example is “Taken together, these structural and computational analyses show how the broadly selective SLBR Hsa uses both steric and electrostatic interactions to exclude specific structural additions to the glycan ligands”

Page 4 last paragraph: While I agree the substitution of the CD and FG loops decreases binding to specific sialoglycans – it should be noted that their substitution also increases binding to sTa for UB10712

In the introduction the Authors state that binding to sTa with high affinity correlates with pathogenicity in endocardial infections – There is a lot of diversity in glycan structures across species and I think it is important for the text to indicate these studies were in an invivo animal model not in humans.

Minor concerns for clarity:

“V-set Ig fold” – to be honest I don't know what the V-set refers to. A little more explanation might help the non-structural reader.

“Because they may sample the ligand bound form even in the absence of sialoglycan, this suggests a conformational selection mechanism.” This is an easy concept, but not a term I was familiar with. A sentence of two about what a conformational selection mechanism is may would make it clearer for some of your targeted audience

Page 3: I don't see anywhere that the PhiTRX motif is identified as being required for activity – it seems to come out of nowhere.

Page 3: MD simulations – MD is not defined until the methods.

Page 3: last paragraph I believe you intended to refer to Fig. 5C

Page 4: It would be useful to name the alpha1,3-fucosylated and O-sulfated glycans to avoid the reader having to revisit figure 1.

Page 5: I think it would be clearer if the striking difference was stated.

Page 5: Where is the evidence presented that L442Y and Y443N etc. directly contact the ligand? – can that information be added?

Page 6: I agree existing data indicates the major ligand bound in the oral cavity is MUC7, but is it really known that all SLBRs bind this mucin? Might SLBRs with different binding specificities bind other mucins or oral glycoproteins?

Perhaps it would be more accurate to narrow this statement somehow –

It is not clear why the structures in Figures 1 and 2 don't use the same format.

Figure 5: in C it doesn't seem the bound structure is shown in light green.

Figure 6: It is not clear what is meant by in each case, sTa binding increases. Do you mean for the chimera's or as the concentration of glycan is increased? I assume the former.....

Figure 8. Panel D is not called out in the legend.

Table S4: This table confuses me. What experiments were used to calculate these was it those in figure 6? Perhaps a little more info in the legend about the ELISAs would increase clarity

Fig. S2 the ion is not defined as such in the legend.

Fig. S6B – it is not clear why the legend defines this as GspBsiglec and not SLBRgspB (also occurs in one place in the text). Also, are the error bars shown in gray?

Reviewer #2 (Remarks to the Author):

The author's have provided in depth analysis of high-quality crystal structures of several SLBR's. These were used to identify receptor binding regions, which were then demonstrated to be interchangeable. Single point mutations can drive ligand affinity by several orders of magnitude. This help explain how certain bacteria can use these SLBRs to rapidly adapt to changing environments and is potentially a factor in the switch from commensalism to pathogenicity. Importantly, they have determined the structures of five common sialo-glycans bound to SLBR-Hsa. This provides a deeper understanding of the specificity and variability of these common bacterial modules, and how they relate to commensalism and pathogenicity. In the end the author's have provided a model for how the three main loops of SLBR's are used to control different aspects of glycan affinity, such as length, sulfation, and breadth of specifcity. This will be of use to the broader community of streptococcal researchers and specifically for those with an interest in the important host-bacteria interactome. I recommend this study for publication with minor revisions.

Comments

The authors demonstrate that the SLBRs GspB and SK150 do not easily alter their binding spectrum, while Hsa, SK678, UB10712 readily does so. This correlate with the phylogenetic grouping in Fig 2, but is not discussed further. Seeing as phylogeny was highlighted by the authors as a poor predictor of cognate ligands it leaves me wondering if the authors missed an important discussion point here.

The glycan sLeA is a notable absence from the study despite it being being the natural extension of testing sLeC. Can the authors could comment on that. Perhaps it was tested in the extended glycan array?

Page 4

The structure suggests that a cation, tentatively assigned as Na⁺, binds near this site to help bridge the interaction.

What evidence is there for assigning a sodium to this location over other ions such as calcium, magnesium, or even water? The structure's resolution is 2.5Å so any cation placement that is brought up for discussion should be appropriately justified. It is noted that "the coordination geometry is distorted" but no further explanation for the reason(s) why a cation in this location is still warranted. The references Fig 4D, and Fig S8F shows the placement of the sodium atom but does not convey the interactions it is making with its surroundings, leaving me doubting its validity.

Page 4 & Fig S8

Differences in the SLBR ligand interactions predominantly map to the variable third sugar of the glycan (Fig. S8B-S8E).

I assume it was intended to reference S8C-S8F, yet it is not clear to me what the authors refer to by this. Apart from sTa where E298 contact the GalNAc nitrogen, the variable third sugar of the glycan is not shown to interact with anything in these figures. Are there more interactions not depicted? Can you clarify these differences?

Minor comments

Table S4.

The EC50 aquired from ELISA-data by linear regression is shown without without error estimation. Please include the confidence intervals for the EC50 values

Page 3

...SLBRGspN-Siglec...

SLBRGspB-Siglec

Page 3

but little detectable binding to any of the tetrasaccharides

Please list the tested tetrasaccharides by here. It is not obvious which (or how many) saccharides are tested.

Fig S5

Cropped to much on the left

Fig S8

There is a stray "6" on top of the E298 backbone in panel C.

Reviewer #3 (Remarks to the Author):

It is known that streptococci and staph have adhesins with Siglec-like binding regions (SLBRs) that recognize the sialic acid-containing epitopes Neu5Aca2-3Gal that occur in larger glycans and are found in mucins and other glycoproteins. In this manuscript the authors explore the structures of 5 representative SLBRs from and explore the molecular basis of receptor binding to such sialic acid ligands.

The strength of this manuscript is in the molecular structures of the SLBRs, and the evidence that alterations in structure can drive changes binding to sialic acid-containing ligands. This in itself is important and interesting. The insights into molecular binding to such glycans is important to the field. However, a key weakness of the study is that the physiological and molecular interactions are not well described here. The novelty of the manuscript is clearly in the structural studies and not really on the physiological relevance side, but understanding the structures of these SLBRs and their interactions with glycans is important.

Unfortunately, the study does not provide clear insights into the biological nature of these interactions with physiological receptors, despite studies of binding to plasma and salivary glycoproteins, and does not really make any predictions about that either. This reviewer realizes that may be a lot to ask in this type of study, as the focus is more on the structure than the function. The glycans on physiological glycoproteins recognized by the SLBRs and chimeras are not clear. No doubt, the study is interesting insofar as it helps to define amino acid residues within the SLBRs that control receptor specificity, but there is little biological validation provided.

The sialic acid-glycan array analyses are interesting and of course indicate differential specificity, but it is not clear that the structural studies in the paper on glycoprotein glycans that are fucosylated and sulfated and sialylated, for example, actually explain all the binding data from those arrays. Much of the binding information is already available in a way, so while this information is interesting, it is not unexpected.

A key finding is that changes in binding specificity correlates not just with the SLBRs, but to is the loops that surround the ligand-binding pocket. In this regard, however, the crystal structures of these chimeras is not presented so that the molecular explanation for this change is not clear, other than a prediction that the flexibility of these loops affect binding.

The key contribution of this manuscript is the crystal structures of these 5 SLBRs, but the crystals of only two of these allowed co-crystallization with sialic acid-containing ligand.

Another problem, with the study and its conclusions is that glycans they have identified occur on glycoproteins (and perhaps glycolipids, which are not discussed), and the conformational contributions of such glyco-protein interactions are not explored. The binding data to plasma glycoproteins is rather preliminary and simply shows interactions without exploration of the molecular basis of such binding. The relatively preliminary and somewhat phenomenological nature of binding to plasma and salivary glycoproteins, as illustrated in Fig 9, only indicates that structural changes in the SLBRs alter their affinities (which are not discussed nor evaluated), and/or their specificities, but the connection of such changes to the actual glycan structures on plasma and salivary glycoproteins is not clear. Different donors for the salivary samples give different results, suggesting that glycosylation differences in the salivary mucins drive changes in binding, but again the details of such changes are not clear.

The connection of glycan array data to endogenous ligand structures on glycoprotein ligands is not really clear, and the data suggests that these are not strictly correlated. The plasma glycoprotein glycans recognized by SLBRs are not defined, and the salivary mucin, e.g. MUC7 glycans, are also

incredibly varied and the results seems rather phenomenological in regard to whether mucin is bound or not.

The binding affinities are discussed in terms of highest affinity or decreased affinity, but the actual affinities to glycans are not described for any of the SLBRs. It would seem straightforward, given that some of the sialic acid-containing glycans are in hand, to conduct microcalorimetry or other studies on the binding affinities.

Minor points:

The authors state their intention to identify human glycan receptors, but the sialic acid-containing glycans they predict as receptors are probably found in most mammals. Thus, it might be worthwhile to mention this as oral colonization might not be human specific.

The linkage of the different sialic acid-containing epitopes to the underlying glycan structure are likely to greatly influence binding, especially in the context of the intact glycoproteins, but this needs to be discussed.

REVIEWER COMMENTS

Reviewer #1 (Remarks to the Author):

Review for Bensing et al. Submitted to Nature Communications

As a bacterial geneticist with an interest in streptococcal carbohydrate binding proteins I am not qualified to comment on the appropriateness of the structural, or some of the biochemical, methods and much of the data analysis. I limit my review to the biological relevance, impact on the field, and the ability of a highly interested individual with different expertise to comprehend the data.

Siglec-like domain containing proteins are important virulence factors in many streptococci that colonize the oral cavity and can cause infective endocarditis. Their role in the oral cavity is less defined, but they can bind sialylated structures on mucins and are likely critical in that environment too. These Siglec-like domain containing proteins all bind sialic acid but can have different binding specificities. Despite structural analysis of some Siglec-like domains we are currently unable to determine binding specificity without a glycan array and even then, the mechanism which defines binding specificity was unclear. **This manuscript transforms our understanding of the binding mechanism.** It has been proposed that three loops in the Siglec-like domains that contribute to sialic acid binding, but this is the first time that clear evidence has been presented. An extensive series of structural and biochemical studies puts together a compelling story that elucidates the roles for

these three loops in the binding specificity of the different Siglec-like domains. The Authors demonstrate that exchange of loops or specific point mutations can alter binding specificity. Furthermore, they examine how these differences altered binding to salivary and serum proteins. Although the diversity of Siglec-like domains in individual species is poorly understood these experiments suggest that strains colonizing different individuals may be selected for by the glycan profile on salivary proteins and that some individuals may be more prone to developing infective endocarditis. The studies included in this manuscript not only shed light on the binding mechanism of these important proteins, but also open the door to development of a panel of specific binding Siglec-like binding proteins that could be used as a tool to determine the presence of specific carbohydrate structures. Glycobiology is currently a difficult field of study due to the lack of tools, but it is a critical area that will be more widely studied in the future.

I believe this manuscript will appeal to a broad range of scientists including those studying colonization and pathogenic mechanisms of streptococci, those studying adhesins and sialic acid binding proteins, glycobiologists, and structural biologists that might well find parallels in other systems.

In general, the manuscript is very well written, and a great deal of thought has gone into the organization of the figures to make a very complex set of analyses understandable by a broad audience. To be honest, it is still pretty dense, but this is difficult to avoid while covering such a large scope. Some additional experiments are likely required and there are no doubt places where this Reviewer feels that clarity could be increased, but there is no doubt this was a massive undertaking that significantly moves the field forward.

This was a thoughtful and constructive review. The reviewer makes a number of excellent points regarding the biology of streptococcal and SLBR binding, and we have incorporated these into our modifications. We thank the reviewer for their overall enthusiasm for the work as well as the specific comments. These comments helped us to better explain our experimental rationale in many places, which we hope will make this manuscript more approachable to a larger readership.

Despite the strength of the manuscript there are some concerns

Data analysis and resulting interpretation and conclusions.

1) There is essentially no statistical analysis of any differences identified in the manuscript. Furthermore, in some cases it is not clear why $n=2$ and whether those were replicates in the same experiments or independent experiments. There would be more confidence in the data if three independent experiments were performed. This also complicates the Authors requests that we compare what appears to be across experiments to determine the effect of chimeras and point mutations. Maybe they were conducted at the same time but if not, it is critical to know that experiments were reproducible. The lack of a clear way to define what constitutes a difference is important and particularly so where I did not see the difference described by the Authors.

We thank the reviewer for this comment. The omission of errors in our supplementary table was unintentional and this has been updated. For the revised version of the manuscript, we have collected additional replicates when needed so that there is a minimum of $n=3$ for all measurements. In addition, we have shown individual data points on all bar graphs, calculated standard deviations, and shown p values for comparators. In some cases, we provided an additional layout of the data so that it is easier to make the requested comparisons.

Going through the specific comments, we appreciate that the reviewer caught several important typos. In a small number of cases, this made small parts of the text incorrect, although the global conclusions were accurate.

Examples are given below:

Page 5: The Authors state “In contrast to the results observed with SLBR Hsa and its close homologs, substitution of the EF loop of SLBR SK150 into SLBR GspB had little impact”. However, I do not see much impact of the EF switch for Hsa binding to 3’sLn – in Fig. S9. Both chimeras and WT proteins bind this carbohydrate well. Also, the Author’s state “in all remaining chimeras, there was substantially decreased binding to both sTa and 3’sLn (Fig. S9C)”. Is this true? the WT binds poorly and so do the chimeras.

We appreciate the Reviewer’s close attention to detail. The reviewer is correct that the EF switch of the Hsa loop into SK678 or UB10712 does not affect binding to 3’sLn. We had not meant to state or imply otherwise. After considering the origins of this comment, we made two changes to clarify the text. First, we provide an additional supplementary figure (**Fig. S15**) that organizes the data by glycan type rather than by mutant and performed statistical analyses. In this layout, it is easier to see that the chimera with the Hsa EF loop has a statistically significant increase in binding to many of the glycans as compared with the parent SLBR_{SK678}.

In addition, we modified the main text to be more specific about how ligand binding is affected by loop replacements. The original text on p.4 stated that broader selectivity occurred via “gain-of-function as reflected in an approximate doubling in binding low-affinity ligands without concomitant loss of binding to preferred ligands”. We have now expanded this on p. 4.

For the second comment regarding the decrease of binding between the SK150-GspB chimeras, the reviewer is correct that the WT and the chimeras both binding poorly to 3’sLn. We have revised the text on p. 5 stating that “in all remaining chimeras, there was little detectable binding to ligands, with substantially decreased binding to sTa and

unchanged, low binding to 3'sLn (Fig. S15C)."

Fig. S9 There was a moderate decrease in SLBR UB10712 Hsa-FG-loop binding to sLeC does not appear to match the data. Furthermore, the legend includes several statements about small differences which it is difficult to determine whether they are significant.

We appreciate the reviewer catching this, which was a typo on our part. The sentence now reads "SLBR_{UB10712}^{Hsa-FG-loop} exhibited a moderate decrease in affinity to sLe^X and 6S-sLe^X and a moderate increase of affinity to sTa and sLe^C." As described in the global comments, we have now laid the data out differently and calculated statistical significance on these comparisons.

The legend of figure 8 states "Both variants had increased binding to 6S-sLeX and 3'sLn and decreased binding to sTa and sLeX" is that actually true? I don't see a reduction for the variants in binding to sLeX.

We appreciate the reviewer catching this, which was a typo on our part. The sentence now reads "Both variants had increased binding to 6S-sLeX, 3'sLn, and sLeC and decreased binding to sTa."

The legend of figure 7 states "When compared to wild-type (see Fig. 6A,B), the GST-SLBR SK678 E298R and GST-SLBR UB10712 E285R variants exhibit increased binding to 6S-sLeX, and reduced binding to 3'sLn and sLeX" Is this true? I don't see much of a difference for SK678 binding to sLeX.

We appreciate the reviewer's attention to detail. To make the changes in ligand binding more apparent, we now added bar graphs as three new panels in Figure 7 (Fig. 7 D-F). These make side-by-side comparisons of parent versus mutant binding to the specified ligands.

2) Much of the data is based on chimeras how is it known that substitution of loops doesn't prevents proper protein folding? I get they are loops, but can this be ruled out? This possibility is easy to eliminate in the case that some binding is increased but for some of the chimeras, such as many of the GspB-SK150 chimeras, no binding was observed. If this is a possibility it should be more clearly acknowledged, and conclusions tempered. Could proper folding be demonstrated by CD or other methodology?

We agree that misfolding is always a possibility. Our past work has shown that misfolded variants of these SLBRs are rare, but when misfolding occurs, it is associated with a distinctive degradation pattern and anomalous migration by size exclusion chromatography. To rule out misfolding as the underlying cause of loss-of-binding, we performed size exclusion chromatography on non-functional mutants. Surprisingly, we found that that the migration of four of the non-binding mutants is consistent with minor folding deficits, i.e. some folded protein was retained, but the peak height corresponding to folded protein was significantly lower and there was the appearance of breakdown products and aggregates. We have noted this at the end of the associated main text paragraph on p. 5 and added new supplemental figure panels, Fig. S16A-D and S19C to demonstrate which variants were potentially mis-folded.

All other variants either bound to ligands robustly (and thus must be folded as they are functional) or exhibited monodispersity consistent with folded protein.

3) Blots. It is hard to see the small difference in size of protein bound by the different Siglec-like domains in 9B. To support the claim they are different, it would seem important to run the samples on the same gel. Furthermore, there is no definitive evidence presented that these proteins are binding MUC7 or the plasma proteins referenced – cited work relies on a correlation of size to previous studies – can the Authors really exclude the possibility of other sialylated

proteins running at these sizes in these donors? Can the Authors please confirm binding is to specific proteins if they wish to draw these conclusions. Or perhaps it doesn't matter, and it can just be hypothesized due to previously reported sizes. The Authors certainly show a difference in binding patterns when loops are switched.

We thank the reviewer for this comment. In terms of the comment about the samples being run on the same gel in Figure 9B, we can confirm that samples probed with WT versus variant BRs were indeed run on the same gel and transferred to the same nitrocellulose membrane. The membrane was subsequently cut in order to be probed separately. We have now clarified this in the legend. A dashed red line has been added across each blot at the 150 kDa mark in order to more clearly see the differences in apparent MW of the MUC7 glycoforms.

Regarding the identity of the glycoprotein ligand(s), both published reports (Prakobphol A, *et al.* Human low-molecular-weight salivary mucin expresses the sialyl lewisx determinant and has L-selectin ligand activity. *Biochemistry* **37**, 4916-4927 (1998); Karlsson NG, Thomsson KA. Salivary MUC7 is a major carrier of blood group I type O-linked oligosaccharides serving as the scaffold for sialyl Lewis x. *Glycobiology* **19**, 288-300 (2009)) and our own findings included here (now summarized in a new Supplementary Table, Table S6) indicate that MUC7 is the most abundant glycoprotein in the 150 kDa region. Minor amounts of MUC5B (>10 mDa) and gp340 (a.k.a. DMBT1; 340 kDa) or fragments thereof, were also detected in the 140-160 kDa region, but the former has larger, more extensively branched, and less sialylated O-glycans, and the latter is primarily N-glycosylated. Thus, neither of those are likely to be direct targets for the SLBRs. To clarify this in the text, we have now added a new supplemental Table (Table S6) that shows mass spectrometry identification of proteins in the 150 kD region.

Conclusions that are unclear and could be clarified

I think the conclusions/suggestions around evolution lacks some clarity. I understand space is limited, but I think this information needs to be clarified and more clearly presented. These bacteria are naturally transformable and hence likely undergo frequent horizontal gene transfer. They also exist in an environment where many other strains of oral streptococci that also contain Siglec-like domains are present making genetic recombination more likely. I think the fact that there are more changes in the loops observed as these regions can tolerate more differences should be included in the discussion. Some of these changes will alter the binding range and perhaps enable the bacterium to bind a different sialic acid structure, which could allow colonization of a different region of the oral cavity (I believe there is some evidence glycans are different in saliva at different sites in the oral cavity) or binding to different salivary components, or different individuals in the population. There might even be the opportunity to bind different sialylated bacteria (theoretically at least). It is not clear what the Authors mean by new anatomical location or in a new host— are you suggesting different animals or body sites – or the narrower ideas given above. It is also not clear what is meant by “it may also favor pure commensalism versus pathogenesis”. The selective pressure for evolution of these species is in the mouth – I don't believe there is any selective pressure for or against bacteria binding to platelets or the heart valve. Perhaps that sentence can be clarified.

We thank the reviewer for these thoughtful comments. We agree with the points raised. In particular, the endovascular environment is not the natural habitat for oral streptococci and thus interactions such as platelet binding are not likely to affect selective pressure. We have revised our discussion to better reflect this.

Representation of the field

In the introduction the Authors state that we do not know what dictates whether these bacteria can switch from commensal to pathogen, but do we even know that there are oral streptococci within these species that are unable to become pathogens in humans? These studies are needed, and it is certainly likely that some species can cause disease

and others cannot, but I don't of any studies that address this. Can more information be given or the sentence altered to more accurately reflect the understanding in the field?

The reviewer raises a number of good points, for which the answers are complex and not entirely conclusive. It is not entirely clear how much species and strains of oral streptococci vary in their ability to produce endocarditis. We know from clinical studies that strains of *S. sanguinis*, *S. gordonii*, *S. mitis*, and *S. oralis* are the most common causes of streptococcal endocarditis (Douglas, C.W., et al. (1993). Identity of viridans streptococci isolated from cases of infective endocarditis. *Journal of Medical Microbiology* 39, 179-182; Isaksson, J., et al (2015). Comparison of species identification of endocarditis associated viridans streptococci using rnpB genotyping and 2 MALDI-TOF systems. *Diagn Microbiol Infect Dis* 81, 240-245.) accounting for a significantly higher percentage of cases than other oral species. In addition, patients with bacteremia by these four species are more likely have endocarditis (Chamat-Hedemand *et al.* (2020). Prevalence of infective endocarditis in streptococcal bloodstream infections is dependent on streptococcal species. *Circulation* 142(8):720-730), indicating that these species are more capable of producing endocardial infection, when in the bloodstream. A number of virulence factors have been identified in animal models of endocarditis, where mutagenesis of targeted genes in the above species reduces virulence. This suggests that natural strains lacking these virulence factors would be less likely to produce endocarditis. Since at least some of these virulence factors are not found in all oral streptococci, it could mean these deficient strains or species are less virulent in the endovascular environment.

However, we are not aware of any studies that directly test whether endocarditis-associated isolates are more capable of producing endocarditis, as compared with oral isolates. Moreover, genomic studies of oral versus endocarditis strains of *S. sanguinis* and *S. gordonii* have had limited success in identifying genes that are over-represented in endocarditis isolates (Iversen *et al.* (2020). Similar genomic patterns of clinical infective endocarditis and oral isolates of *Streptococcus sanguinis* and *Streptococcus gordonii*. *Scientific reports* 10, 2728.)

These findings may reflect, 1) redundant virulence factors (e.g., multiple adhesins) or 2) the endocarditis-associated genes are a subset of genes needed for survival within the oral cavity. Consistent with the latter hypothesis, we and others have found that all strains of *S. sanguinis* and *S. gordonii* encode Siglec-like adhesins, presumably because they are required for oral colonization. In addition, our studies using a well-established animal model of endocarditis have shown that the type of sialoglycan bound by these organism affects virulence, indicating that in fact, strains may vary in their ability to produce human disease, depending on which Siglec they express.

Although we cannot address these points in depth within the introduction, we have revised this section to better reflect the points raised by the reviewer and the above concepts.

Other concerns

In some of sections of the paper it reads to this Reviewer as if the bacteria are making conscious decisions. For example, around evolution – it reads that it was a decision on the part of the bacteria not random events that were maintained if they provided a selective advantage. Another example is “Taken together, these structural and computational analyses show how the broadly selective SLBR Hsa uses both steric and electrostatic interactions to exclude specific structural additions to the glycan ligands”

The specific sentence has been modified and we have done our best to identify other statements that might imply sentence on the part of bacteria or proteins.

Page 4 last paragraph: While I agree the substitution of the CD and FG loops decreases binding to specific sialoglycans – it should be noted that their substitution also increases binding to sTa for UB10712

Now noted in text

In the introduction the Authors state that binding to sTa with high affinity correlates with pathogenicity in endocardial infections – There is a lot of diversity in glycan structures across species and I think it is important for the text to indicate these studies were in an in vivo animal model not in humans.

In response to the reviewer's concerns, we have reviewed the relevant sections of the paper to ensure that all data from animal studies are clearly labeled as such. We agree that animals can differ as to the specific glycan modifications on proteins, and this difference can limit the applicability of data from animal models to human infection. In the case of endocarditis, studies in animals (mostly rats, but also rabbits and mice) indicate that these models are relevant to human disease because 1) binding of streptococci to both human and rat platelets, which is a key interaction in the pathogenesis of infective endocarditis, is dependent on the expression of the Siglec-like adhesins, for both animal and human platelets; 2) For both rats and humans, the platelet receptor appears to be GPIb α ; and 3) MS studies indicate the sialoglycans on GPIb α from humans and rats are similar. Unfortunately, more directly corroborative data from human cases of endocarditis are not available. In addition, the role of sialoglycan binding has not been examined in clinical studies, and there are no large clinical databases that might provide such information.

Minor concerns for clarity:

“V-set Ig fold” – to be honest I don't know what the V-set refers to. A little more explanation might help the non-structural reader.

On first use of the term “V-set Ig fold”, we now indicate that the V-set Ig fold contains predominantly beta secondary structural elements and was named for its discovery as the fold of antibody variable domains. It is known for being a particularly stable protein fold that is readily amenable to substitution in the antigen-recognition region.

“Because they may sample the ligand bound form even in the absence of sialoglycan, this suggests a conformational selection mechanism.” This is an easy concept, but not a term I was familiar with. A sentence of two about what a conformational selection mechanism is may would make it clearer for some of your targeted audience

On p. 3, we have now expanded on the concept of conformational selection and specifically defined it as a situation where the structural change of the binding protein precedes binding of ligand. The ligand then selects and stabilizes this high-energy conformation. We further indicate that conformational selection of a binding protein has been shown to correlate with broad selectivity and provide citations.

Page 3: I don't see anywhere that the PhiTRX motif is identified as being required for activity – it seems to come out of nowhere.

We appreciate the reviewer pointing this out, the definition was unintentionally omitted from the introduction. This has now been defined on p. 2 with a note that past mutagenesis of the PhiTRX motif shows that it is critical for ligand binding. We further include the references to past work that defined this motif.

Page 3: MD simulations – MD is not defined until the methods.

MD is now defined as molecular dynamics on page 3.

Page 3: last paragraph I believe you intended to refer to Fig. 5C

Thank you for catching this, it has been corrected.

Page 4: It would be useful to name the alpha1,3-fucosylated and O-sulfated glycans to avoid the reader having to revisit figure 1.

The text has been changed to, “In considering how the α 1,3-fucose in glycans **such as sLe^x and 6S-sLe^x...**”

Page 5: I think it would be clearer if the striking difference was stated.

The sentence now reads: “The ready ability of the three SLBR_{Hsa}-like adhesins to change their glycan binding profile contrasted strikingly with the inability of the two SLBR_{GspB}-like adhesins to similarly change their preferred ligand. This might be explained in several ways...”

Page 5: Where is the evidence presented that L442Y and Y443N etc. directly contact the ligand? – can that information be added?

We thank the reviewer for this comment – the rationale for the design of these variants was unintentionally omitted. The structure of SLBR-GspB bound to sTa bound, reported in this manuscript, identifies that atoms from residues L442 and Y443 closely approach atoms in the third sugar of sTa, i.e. GalNAc.

To address this comment, we have now explained the rationale behind the mini-chimera design and have include a figure panel that shows the locations of these residues, **Fig. S17**. We have additionally modified the text on p.5 to read “To better understand why SLBR_{Hsa}-like proteins were more mutable, we leveraged our crystal structure of SLBR_{GspB} in complex with sTa (**Fig. 4E**) and identified that SLBR_{GspB}^{L442} and SLBR_{GspB}^{Y443} closely approach the GalNAc (**Fig. S17A,B**). We engineered SLBR_{GspB-SK150} “mini-chimeras” that swapped single amino acids at these positions with the equivalent residues from SLBR_{SK150}.”

Page 6: I agree existing data indicates the major ligand bound in the oral cavity is MUC7, but is it really known that all SLBRs bind this mucin? Might SLBRs with different binding specificities bind other mucins or oral glycoproteins?

All SLBRs characterized to date bind α 2-3 sialoglycans, and the primary or exclusive salivary ligand is MUC7. Some broader-specificity SLBRs such as SLBR_{Hsa} may have a relatively weak affinity for other sialylated salivary glycoproteins, but we have shown that MUC7 is by far the predominant ligand detected by Far western blotting and affinity capture.

It is not clear why the structures in Figures 1 and 2 don't use the same format.

We have modified the glycan structures to be consistent in figures 1 and 2.

Figure 5: in C it doesn't seem the bound structure is shown in light green.

In panel C, the EF loop position from the bound structure is shown in light green. The glycan and the rest of the SLBR are shown in black and grey respectively. We wonder if the color is not reproducing well in the reviewer's PDF or print out? If this is not reproducing for all readers, we can modify the text if we know what this color looks like to you.

Figure 6: It is not clear what is meant by in each case, sTa binding increases. Do you mean for the chimera's or as the concentration of glycan is increased? I assume the former.....

The text was intending to indicate that each of these chimeras binds sTa more strongly than does the parent adhesin. We have updated the figure legend accordingly.

Figure 8. Panel D is not called out in the legend.

Corrected, thank you.

Table S4: This table confuses me. What experiments were used to calculate these was it those in figure 6? Perhaps a little more info in the legend about the ELISAs would increase clarity

The reviewer is correct that the values were calculated from the ELISA measurements across Figs. 6, 7, 8, S1, S15, S17, and S19. The table legend has been updated for clarity.

Fig. S2 the ion is not defined as such in the legend.

The legend now indicates that the spheres are ions and gives the tentative assignment in each case.

Fig. S6B – it is not clear why the legend defines this as GspBsiglec and not SLBRgspB (also occurs in one place in the text). Also, are the error bars shown in gray?

We have corrected the SLBR nomenclature errors. The error bars are shown in thin black lines and the grey shaded regions are the CD, EF, and FG loops. The figure legend has been updated for clarity.

Reviewer #2 (Remarks to the Author):

The author's have provided in depth analysis of high-quality crystal structures of several SLBR's. These were used to identify receptor binding regions, which were then demonstrated to be interchangeable. Single point mutations can drive ligand affinity by several orders of magnitude. This help explain how certain bacteria can use these SLBRs to rapidly adapt to changing environments and is potentially a factor in the switch from commensalism to pathogenicity. Importantly, they have determined the structures of five common sialo-glycans bound to SLBR-Hsa. This provides a deeper understanding of the specificity and variability of these common bacterial modules, and how they relate to commensalism and pathogenicity. In the end the author's have provided a model for how the three main loops of SLBR's are used to control different aspects of glycan affinity, such as length, sulfation, and breadth of specificity. This will be of use to the broader community of streptococcal researchers and specifically for those with an interest in the important host-bacteria interactome. I recommend this

study for publication with minor revisions.

Comments

The authors demonstrate that the SLBRs GspB and SK150 do not easily alter their binding spectrum, while Hsa, SK678, UB10712 readily does so. This correlate with the phylogenetic grouping in Fig 2, but is not discussed further. Seeing as phylogeny was highlighted by the authors as a poor predictor of cognate ligands it leaves me wondering if the authors missed an important discussion point here.

We thank the reviewer for this important comment. It was not entirely clear to us why the related Hsa, SK678, and UB10717 appeared to be more easily mutable than GspB and SK150. It is curious to think about why this could be so, and we postulate several possibilities on p. 5. Given the small number of GspB-like SLBRs in this study, we are somewhat hesitant to expand this discussion without substantial additional work that would be beyond the scope of this report.

The glycan sLeA is a notable absence from the study despite it being the natural extension of testing sLeC. Can the authors could comment on that. Perhaps it was tested in the extended glycan array?

sLeA is included in the extended glycan array presented here (CFG glycan array version 5.4) but was not specifically mentioned in text because it is a non-binding ligand. We tested binding of wildtype SLBR_{SK1}, SLBR_{SK678}, SLBR_{Hsa}, and SLBR_{UB10712} to sLeA previously (Fig 7B in Bensing et al Glycobiology **26**(11): 1222-1234. 2016) where it was also found to be a non-binding ligand. In this paper, sLe^A is tested for binding to SLBR_{SK150} (Fig S1B) via an ELISA. Taken together, sLeA has never been a high affinity ligand to the SLBRs that we have tested so far, which is why it is only discussed minimally. To clarify in the revision that sLeA was tested, we now add a new table, **Supplemental Table 5**, that indicates the ligands included in the CFG arrays. Because of the size of this table, it is uploaded separately from the rest of the SI.

Page 4

The structure suggests that a cation, tentatively assigned as Na⁺, binds near this site to help bridge the interaction. What evidence is there for assigning a sodium to this location over other ions such as calcium, magnesium, or even water? The structure's resolution is 2.5Å so any cation placement that is brought up for discussion should be appropriately justified. It is noted that "the coordination geometry is distorted" but no further explanation for the reason(s) why a cation in this location is still warranted. The references Fig 4D, and Fig S8F shows the placement of the sodium atom but does not convey the interactions it is making with its surroundings, leaving me doubting its validity.

The reviewer brings up a thoughtful point about solvent assignment. The identity of solvent-derived molecules cannot be unambiguously determined in X-ray diffraction studies. As a result, cations (as well as most biological anions and other solvent molecules) must always be referred to as tentatively assigned. The composition of the purification and crystallization buffers, and the coordination sphere generally allows only a best guess at the assignment. In this case, we assigned this density as a positive charge because it interacted with partially negatively-charged ligands (O8 and O9 of sialic acid, two oxygens on the sulfate, and oxygens on an alternative conformation of the Glu 286 side chain) and no delta-positive ligands. We assigned it as a Na⁺ based upon the prevalence of Na⁺ in the crystallization condition and on the density being consistent with the number of electrons in Na⁺.

Based on the reviewer interest in the site, we re-evaluated the electron density with simulated annealing omit maps. From this, we speculate that the species could be a mixture of Na⁺ and glycine, which is part of the crystallization buffer conditions. As a result, we have changed the assignment to the PDB designation of unknown ligand (UNL) and modified the text accordingly.

No matter the identity of this solvent molecule, it bridges the 6S sulfation of 6S-sLeX with O8 and O9 of sialic acid and the Glu 286 side chain of SLBR_{Hsa}. The local concentration of negative charge helps to explain why ligands bearing 6S sulfation are not high affinity ligands.

Page 4 & Fig S8

Differences in the SLBR ligand interactions predominantly map to the variable third sugar of the glycan (Fig. S8B-S8E). I assume it was intended to reference S8C-S8F, yet it is not clear to me what the authors refer to by this. Apart from sTa where E298 contact the GalNAc nitrogen, the variable third sugar of the glycan is not shown to interact with anything in these figures. Are there more interactions not depicted? Can you clarify these differences?

Thank you for pointing this out, this has been changed in the text, now Fig. S13C-S13F based on the addition of new supplementary figures.

In terms of specific interactions to the third sugar, SLBR_{Hsa}^{D356} forms a hydrogen bond to third sugar of sTa, which is shown in a revised version of Fig. S8C. Remaining interactions are van der Waals or electrostatic. This is consistent with previous work from our lab showing that SLBR_{SrPA} makes contacts to the third sugar via pi-stacking of the glycan with a phenylalanine rather than via hydrogen-bonding interactions (see Fig. 7 in Loukachevitch et al. Biochemistry. 2016 Oct 25;55(42):5927-5937. doi: 10.1021/acs.biochem.6b00704). These types of interactions are difficult to depict in crystallographic figures and were therefore only discussed in the text. To now illustrate the interactions for the reader, we used the program LigPlot to depict interactions to four ligands as four new supporting figures, **Fig. S3 – S7**.

Minor comments

Table S4.

The EC50 acquired from ELISA-data by linear regression is shown without without error estimation. Please include the confidence intervals for the EC50 values

Thank you. This was unintentionally omitted and has now been corrected.

Page 3

...SLBRGspN-Siglec...

SLBRGspB-Siglec

Thank you for catching this error, it has been fixed.

Page 3

but little detectable binding to any of the tetrasaccharides

Please list the tested tetrasaccharides by here. It is not obvious which (or how many) saccharides are tested.

The glycans have now been identified as tri- or tetrasaccharides in the main text as well as legend of Fig. S1, which is the accompanying figure. To further clarify, we have also added a call out to Fig 1, which contains the glycan structures.

Fig S5

Cropped too much on the left

Corrected. Thank you.

Fig S8

There is a stray “6” on top of the E298 backbone in panel C.

Corrected. Thank you.

Reviewer #3 (Remarks to the Author):

It is known that streptococci and staph have adhesins with Siglec-like binding regions (SLBRs) that recognize the sialic acid-containing epitopes Neu5Aca2-3Gal that occur in larger glycans and are found in mucins and other glycoproteins. In this manuscript the authors explore the structures of 5 representative SLBRs from and explore the molecular basis of receptor binding to such sialic acid ligands.

The strength of this manuscript is in the molecular structures of the SLBRs, and the evidence that alterations in structure can drive changes binding to sialic acid-containing ligands. This in itself is important and interesting. The insights into molecular binding to such glycans is important to the field.

Thank you

However, a key weakness of the study is that the physiological and molecular interactions are not well described here. The novelty of the manuscript is clearly in the structural studies and not really on the physiological relevance side, but understanding the structures of these SLBRs and their interactions with glycans is important.

Unfortunately, the study does not provide clear insights into the biological nature of these interactions with physiological receptors, despite studies of binding to plasma and salivary glycoproteins, and does not really make any predictions about that either. This reviewer realizes that may be a lot to ask in this type of study, as the focus is more on the structure than the function. The glycans on physiological glycoproteins recognized by the SLBRs and chimeras are not clear. No doubt, the study is interesting insofar as it helps to define amino acid residues within the SLBRs that control receptor specificity, but there is little biological validation provided.

The reviewer raises a good point regarding the connection between the (in vitro) molecular and (in protein) physiological interactions. In the revision, we have more explicitly connected the in vitro binding studies with the binding studies in saliva and plasma. As a part of this, we have included two new supplemental tables (**Table S5 and Table S6**) that list out the glycans used in the invitro studies and show the mass spectrometry of the proteins identified in the studies from saliva. The revised text itself now more explicitly indicates that the terminal glycans recognized in saliva and plasma are indeed the trisaccharides that bind in the in vitro ELISA assays.

We do note that more complete answers to the reviewer’s queries are complex and not entirely conclusive. MUC7 appears to be the major glycoprotein ligand in saliva for the Siglec-like adhesins. Binding of streptococci to MUC7 is thought to be important for the initial colonization of the oropharynx, which serves as a basis for the subsequent development of the oral microbiome. The role of MUC7 in oral biology is uncertain, with few studies that address its role in colonization, and in particular, no in vivo work and it has not been evaluated how this might contribute to commensal colonization. It is therefore difficult to make predictions as to the physiologic impact of differences in sialoglycan recognition. Developing experimental *in vitro* and *in vivo* models that would directly test this issue would be well beyond the scope of this paper. However, given that Siglec-like SRR adhesins are ubiquitous in *S. gordonii* and *S. sanguinis*, it is very plausible that these adhesins are essential, and that they contribute to colonization via their interaction with MUC7 immobilized on dental surfaces (Gibbins et al (2014) Concentration of salivary protective proteins

within the bound oral mucosal pellicle. Oral Dis 20, 707-713.; Ruhl et al (2005) Proteins in whole saliva during the first year of infancy. J Dent Res 84, 29-34.) We hypothesize that the variety of SRR binding regions and preferred ligands may reflect the heterogeneity of sialoglycan modifications on MUC7, both within and between human hosts. It's also possible that these adhesins interact with unidentified ligands, although our studies probing human saliva and plasma would indicate otherwise (Bensing, B.A., et al. (2018). Streptococcal Siglec-like Adhesins Recognize Different Subsets of Human Plasma Glycoproteins: Implications for Infective Endocarditis. *Glycobiology*. 28(8): 601–611). In our revised discussion, we address these issues more fully.

We would also note that with regard to the pathogenesis of infective endocarditis, our previous publication using isogenic strains has clearly shown that virulence is significantly affected by the type of sialoglycan bound, through as yet not well-defined mechanisms (see Bensing (2019) PLoS Pathog Jun 24;15(6):e1007896. doi: 10.1371/journal.ppat.1007896). For example, we do know from clinical studies that certain species of oral streptococci (particularly *S. sanguinis*, *S. gordonii*, *S. mitis*, and *S. oralis*) are more likely to produce endocarditis, and that bacteremia by these species is more likely to be due to endocarditis. However, we are not aware of any clinical study indicating that endocarditis-associated isolates of the above four species are more capable of producing endocarditis, as compared with oral isolates of these species. Moreover, genomic studies of *S. sanguinis* and *S. gordonii* have been largely unsuccessful in identifying specific virulence genes that are over-represented in IE strains. As the reviewer suggests, these findings indicate that most strains of these species may be capable of producing IE. Consistent with hypothesis, we have found that all strains of *S. sanguinis* and *S. gordonii* encode Siglec-like adhesins. However, our studies using a well-established animal model of IE have shown that the type of sialoglycan bound by these organism affects virulence, indicating that in fact, strains may vary in their ability to produce human disease, depending on which Siglec they express.

In aggregate, our studies provide significant physiologic insights, both in terms of the importance of the Siglec adhesins in the oral environment, as well as within the endovascular space.

The sialic acid-glycan array analyses are interesting and of course indicate differential specificity, but it is not clear that the structural studies in the paper on glycoprotein glycans that are fucosylated and sulfated and sialylated, for example, actually explain all the binding data from those arrays. Much of the binding information is already available in a way, so while this information is interesting, it is not unexpected.

The reviewer brings up an important point: do these structural properties explain all of the phenotypic properties that we observe, i.e. selectivity in vitro and in the context of human proteins. The short answer is yes. We have now modified the text in the results section to better clarify how the structures inform the selectivity. Toward this, we have included a new supplemental figure (**Fig. S14**) that shows how the hydrogen-bonds of the chimera change with different ligands.

The reviewer also hints at the overarching question that we have been working toward in the laboratory, i.e. can we predict biological behavior (particularly relative virulence) of a strain of streptococci if we sequenced its adhesin? Past work (now better highlighted in the introduction) identifies that sTa binding correlates with virulence in an animal model (see Bensing (2019) PLoS Pathog Jun 24;15(6):e1007896. doi: 10.1371/journal.ppat.1007896). Currently, the most pressing question arises because as more streptococci with SLBRs are discovered, it is increasingly unclear how we predict what sequence binds to sTa versus another ligand.

This study provides a major step toward that overarching goal because it identifies what regions of sequence dictate specificity, which would tell us where to look for virulence determinants. The identification of the specificity determinants as being almost entirely within three loops was quite unexpected. Past work showed that the difference

in binding to animal versus human forms of sialic acid (Neu5Gc versus Nue5Ac) could have complex molecular origins and involve coordinated action of distal amino substitutions that worked together allosterically. This is in striking contrast to what is observed here, where specificity is encoded by localized regions of the protein sequence that are immediately adjacent to the binding site.

A key finding is that changes in binding specificity correlates not just with the SLBRs, but to is the loops that surround the ligand-binding pocket. In this regard, however, the crystal structures of these chimeras is not presented so that the molecular explanation for this change is not clear, other than a prediction that the flexibility of these loops affect binding.

We agree that the key finding is that the loops dictate selectivity. We strongly disagree with the conclusion that a lack of additional crystal structures means that we do not know the molecular basis for this change. Given the >80% sequence identity of the Siglec domain of the parent adhesins, the functional binding patterns of the chimeras, and considering the central dogma of structural biology (that sequence dictates fold), we know that the sialic acid binding motif retains its structure and that the loops retain a similar conformation as is observed in their parent structures. In the same way, a DNA sequence would result in the same amino acid being translated regardless of its context. Additional experimental structures are not necessary to support this conclusion.

Given the reviewer's interest in the chimeric structures, we used computational methods to develop a model of the SLBR_{SK678}^{Hsa-loops} chimera, docked the sTa and 3'sLn ligands in both the parent and chimeric SLBRs, and performed molecular dynamics simulations to evaluate relative binding strength of the ligands. These identify the changes in selectivity result from contacts adjacent to the third sugar, highlighting the residues that we mutated as critical for selectivity. In the revised manuscript, we included this analysis in a new supplemental figure, **Fig. S14**.

The key contribution of this manuscript is the crystal structures of these 5 SLBRs, but the crystals of only two of these allowed co-crystallization with sialic acid-containing ligand.

The reviewer is correct. Crystallization with ligands is commonly not a matter of effort but involves having crystallization conditions that support ligand binding. This is non-trivial. For example, we required five different crystal forms of SLBR_{GspB} (two previously published and three here) to find conditions that supported high-occupancy ligand binding despite a reported nM affinity (Takamatsu, D., et al (2005). Binding of the *Streptococcus gordonii* surface glycoproteins GspB and Hsa to specific carbohydrate structures on platelet membrane glycoprotein Ib α . *Mol Microbiol* 58, 380-392). Fortunately, the sialic acid binding motif anchors the ligand position and allows us to make good inferences about the location of where the ligands bind for the three SLBRs where we do not have experimental costructures. This is in line with the central dogma of structural biology. The mutagenesis that changes selectivity supports this equivalent binding position in all cases.

Another problem, with the study and its conclusions is that glycans they have identified occur on glycoproteins (and perhaps glycolipids, which are not discussed), and the conformational contributions of such glyco-protein interactions are not explored. The binding data to plasma glycoproteins is rather preliminary and simply shows interactions without exploration of the molecular basis of such binding. The relatively preliminary and somewhat phenomenological nature of binding to plasma and salivary glycoproteins, as illustrated in Fig 9, only indicates that structural changes in the SLBRs alter their affinities (which are not discussed nor evaluated), and/or their specificities, but the connection of such changes to the actual glycan structures on plasma and salivary glycoproteins is not clear. Different donors for the

salivary samples give different results, suggesting that glycosylation differences in the salivary mucins drive changes in binding, but again the details of such changes are not clear.

We thank the reviewer for this comment. These SLBRs have previously been identified as binding to glycoproteins because of early work that identified platelet glycoprotein Ib as relevant to virulence in an endocardial setting (Takamatsu, D., et al (2005). Binding of the *Streptococcus gordonii* surface glycoproteins GspB and Hsa to specific carbohydrate structures on platelet membrane glycoprotein Ibalpha. *Mol Microbiol* 58, 380-392), the field has focused on the interactions with glycoproteins. There is no a priori reason that glycolipids could not also be biological targets, but to our knowledge this has never been observed.

In terms of the connection between the binding of these SLBRs to salivary glycoproteins on Western versus the actual structures found in the context of MUC7, this was shown by mass spectrometry, as summarized in **Figs. 9B, S20, S21**, and the newly added **Table S6** (note that these figures are renumbered based on the inclusion of additional supplementary figures). We have now added more detail to the results section that the identified glycans in **Fig. S20** are those that pulled down with the given SLBRs (i.e. affinity capture as described in the methods).

The connection of glycan array data to endogenous ligand structures on glycoprotein ligands is not really clear, and the data suggests that these are not strictly correlated. The plasma glycoprotein glycans recognized by SLBRs are not defined, and the salivary mucin, e.g. MUC7 glycans, are also incredibly varied and the results seems rather phenomenological in regard to whether mucin is bound or not.

We thank the reviewer for this comment. The ligands in the glycan arrays from the National Center for Functional Glycomics were determined as likely physiologically relevant in both humans and animals. Accordingly, the glycan array data show a range of ligands that these SLBRs could potentially bind. The glycan array data show a range of defined ligands that that are bound by these SLBRs in vitro, and that could potentially bind on glycoproteins in vivo. The mass spectrometry in **Fig. S19 and S20** examines the protein-attached sialoglycans that these SLBRs bound in affinity-capture studies using samples from human donors. These are complementary approaches for exploring the possible ligands of these adhesins, and we believe the results are largely concordant.

To clarify this, we have added more detail to the results and added a new supplementary table, **Table S5**, that explicitly indicates the array components. This now better connects the binding of SLBRs to these purified glycans to the binding to terminal trisaccharides from the samples collected from human donors.

The binding affinities are discussed in terms of highest affinity or decreased affinity, but the actual affinities to glycans are not described for any of the SLBRs. It would seem straightforward, given that some of the sialic acid-containing glycans are in hand, to conduct microcalorimetry or other studies on the binding affinities.

In many cases, exact affinity measurements can enhance our understanding of biology. In this case, the goal is to identify preferred ligands and receptors. Measuring affinities would not provide an additional advance or inform on the identity of the preferred receptor. The relative binding shown in the binding curves and the EC₅₀ values reported in **Table S4** fully address the biological question that we are asking.

Minor points:

The authors state their intention to identify human glycan receptors, but the sialic acid-containing glycans they predict

as receptors are probably found in most mammals. Thus, it might be worthwhile to mention this as oral colonization might not be human specific.

Sialoglycan receptors actually differ in humans, mammals, and birds, which is why cross-species transmission is occasional in many cases (and the basis for pandemics). One could consider the low cross-species transmission of the H5N1 bird flu, which binds to sialoglycan receptors. The low rate of species jump reflects differences in the sialoglycan receptors of avian versus human respiratory tract.

Humans also differ from other animals in the types of sialic acid produced. Over the course of evolution, humans have lost the ability to synthesize Sia *N*-glycolylneuraminic acid (Neu5Gc) from *N*-acetylneuraminic acid (Neu5Ac). There is some cross-reactivity of SLBR_{Hsa} and other SLBRs to Neu5Gc, but SLBR_{GspB} is known to bind to the Neu5Ac-containing human glycans selectively in vitro. We direct the reviewer to a large body of literature on differences between human and animal sialic acids; see <https://www.frontiersin.org/articles/10.3389/fimmu.2019.00789/full> for a recent review. To clarify this, we have modified the introduction to point out that the human glycome is known to differ from that of other animals.

Our rationale for this study is to understand how streptococci interact with human receptors, and not the receptors of other animals. We therefore used human receptors and human samples in our studies. To accurately summarize the knowledge of the literature, we carefully evaluated where we used to the word human versus animal and revised for clarity when appropriate. We also included an explicit comment in the introduction on the species differences in glycan repertoire.

The linkage of the different sialic acid-containing epitopes to the underlying glycan structure are likely to greatly influence binding, especially in the context of the intact glycoproteins, but this needs to be discussed.

The reviewer raises a thoughtful point about linkages – both to the protein and inter-glycan linkages. Our unpublished structural work on SLBR_{SrpA} shows that there is not a specific interaction between the SLBR and the protein linkage in that case. Nevertheless, we recognize that protein linkages may be important in other cases and is a focus of ongoing work. A comment to this effect is now included in the discussion.

In terms of inter-glycan linkage, the effect of an altered linkage results in the reorientation of an individual monosaccharide with respect to the body of the glycan. For example, 1-3 versus 1-4 results in rotation of the third sugar of a trisaccharide, but the glycosidic linkage itself does not differ in position between 1-3 and 1-4 linkages and is recognized similarly by the protein (**Fig. S13B**). Instead, differential recognition arises because the protein can accommodate changes in the overall shape of a the different-linked glycan as well as changes the pattern of hydrogen-bonding donors and acceptors. This is illustrated in **Fig. S13B** and the discussion now explicitly includes text related to both glycosidic and protein linkages.

REVIEWERS' COMMENTS

Reviewer #1 (Remarks to the Author):

I appreciate the Authors sincere approach to addressing the concerns raised in my original review. I appreciate that experiments are now an N=3 and that they took care of most of my concerns. However, I still am concerned that the Authors do not know that MUC7 is the protein being bound in saliva. In response to my previous comment the Authors examined proteins present at that position in the gel using mass spectrometry. The most peptides came from MUC7. However, I did not think that this methodology is quantitative. I assume the numbers in columns A and B refer to the numbers of peptides identified from each protein? – there are more from MUC7 but there are relatively large numbers of fragments from other proteins? And does the methodology allow this conclusion? Is it known how much protein would be required for the detection observed? I agree it is likely binding is to MUC7, but I don't see how the Authors demonstrate this is the case. I don't have any easily achievable solutions to this problem except to change the wording in the manuscript to say SLBRs recognized a band consistent with the mobility of MUC7 or similar in a few places.

Based on additional experimentation the Authors acknowledge that some of the chimeras “maybe partially misfolded with significant levels of protein aggregates and break-down products observed”. However, there is no direct acknowledgement that this might be responsible for the lack of binding observed.

Minor point, but the legend for Fig 9 appears to address the panels in the incorrect order.

Reviewer #3 (Remarks to the Author):

In this revised manuscript the authors adequately address a number of issues raised in the initial review. Their combined and revised results provide deeper insights into the structures and potential biological functions of 5 representative (streptococcal Siglec-like adhesins) SLBRs and their interactions with sialic acid ligands.

As structure/function relations are still largely lacking in the field for protein-glycan interactions, the strength of this manuscript is clearly in the molecular structures of the SLBRs, and the evidence that alterations in structure can drive changes binding to sialic acid-containing ligands. The insights into molecular binding to such glycans is important to the field. While a key weakness of the study was felt to be the lack of insight into “physiological” the authors have thoughtfully responded with additional information, and explanations. As this reviewer acknowledges it is certainly not easy in a single study to derive information about such physiological interactions, so the results presented here are clearly a roadmap to such future studies. Again, the important novelty of the manuscript is clearly in the structural studies and understanding the structures of these SLBRs and their molecular interactions with glycans. As such, the studies are rigorous and interesting and help to move the field forward in this important area of host-pathogen interactions.

RESPONSE TO REVIEWER COMMENTS

Reviewer #1 (Remarks to the Author):

I appreciate the Authors sincere approach to addressing the concerns raised in my original review. I appreciate that experiments are now an N=3 and that they took care of most of my concerns. However, I still am concerned that the Authors do not know that MUC7 is the protein being bound in saliva. In response to my previous comment the Authors examined proteins present at that position in the gel using mass spectrometry. The most peptides came from MUC7. However, I did not think that this methodology is quantitative. I assume the numbers in columns A and B refer to the numbers of peptides identified from each protein? – there are more from MUC7 but there are relatively large numbers of fragments from other proteins? And does the methodology allow this conclusion? Is it known how much protein would be required for the detection observed? I agree it is likely binding is to MUC7, but I don't see how the Authors demonstrate this is the case. I don't have any easily achievable solutions to this problem except to change the wording in the manuscript to say SLBRs recognized a band consistent with the mobility of MUC7 or similar in a few places.

Response: We clarified the text to recognize any possible uncertainties, using the suggested wording.

Based on additional experimentation the Authors acknowledge that some of the chimeras “maybe partially misfolded with significant levels of protein aggregates and break-down products observed”. However, there is no direct acknowledgement that this might be responsible for the lack of binding observed.

Response: We clarified on p. 5 that the experiment showed a “size exclusion profile that suggested the presence of breakdown products, indicating that misfolding likely contributes to loss of binding for this variant (Supplementary Fig. 16d).

Minor point, but the legend for Fig 9 appears to address the panels in the incorrect order.

Response: Thank you for noticing. This is corrected.

Reviewer #3 (Remarks to the Author):

In this revised manuscript the authors adequately address a number of issues raised in the initial review. Their combined and revised results provide deeper insights into the structures and potential biological functions of 5 representative (streptococcal Siglec-like adhesins) SLBRs and their interactions with sialic

acid

ligands.

As structure/function relations are still largely lacking in the field for protein-glycan interactions, the strength of this manuscript is clearly in the molecular structures of the SLBRs, and the evidence that alterations in structure can drive changes binding to sialic acid-containing ligands. The insights into molecular binding to such glycans is important to the field. While a key weakness of the study was felt to be the lack of insight into "physiological" the authors have thoughtfully responded with additional information, and explanations. As this reviewer acknowledges it is certainly not easy in a single study to derive information about such physiological interactions, so the results presented here are clearly a roadmap to such future studies. Again, the important novelty of the manuscript is clearly in the structural studies and understanding the structures of these SLBRs and their molecular interactions with glycans. As such, the studies are rigorous and interesting and help to move the field forward in this important area of host-pathogen interactions.

Response: Thank you.